# The intrinsically disordered protein TgIST from Toxoplasma gondii inhibits STAT1 signaling by blocking cofactor recruitment

Zhou Huang[1], Hejun Liu [2,5], Jay Nix [3], Rui Xu[1], Catherine R. Knoverek[4], Gregory R. Bowman [4], Gaya K. Amarasinghe [2] & L. David Sibley [1✉]

Signal transducer and activator of transcription (STAT) proteins communicate from cell-surface receptors to drive transcription of immune response genes. The parasite *Toxoplasma gondii* blocks STAT1-mediated gene expression by secreting the intrinsically disordered protein TgIST that traffics to the host nucleus, binds phosphorylated STAT1 dimers, and occupies nascent transcription sites that unexpectedly remain silenced. Here we define a core region within internal repeats of TgIST that is necessary and sufficient to block STAT1-mediated gene expression. Cellular, biochemical, mutational, and structural data demonstrate that the repeat region of TgIST adopts a helical conformation upon binding to STAT1 dimers. The binding interface is defined by a groove formed from two loops in the STAT1 SH2 domains that reorient during dimerization. TgIST binding to this newly exposed site at the STAT1 dimer interface alters its conformation and prevents the recruitment of co-transcriptional activators, thus defining the mechanism of blocked transcription.

[1] Department of Molecular Microbiology, Washington University School of Medicine, St. Louis, MO 63110, USA. [2] Department of Pathology and Immunology, Washington University School of Medicine, St. Louis, MO 63110, USA. [3] Molecular Biology Consortium, Advanced Light Source, Lawrence Berkeley National Laboratory, Berkeley, CA 94720, USA. [4] Department of Biochemistry and Molecular Biophysics, Washington University School of Medicine, St. Louis, MO 63110, USA. [5] Present address: Department of Integrative Structural and Computational Biology, The Scripps Research Institute, La Jolla, CA 92037, USA. ✉email: sibley@wustl.edu

Interferons (IFN) exert their rapid responses in defense against microbial infection through the signal transducers and activators of transcription (STATs)[1]. Following the binding of interferon to host cell receptors, phosphorylation by Janus kinases recruits STATs via their SH2 domains resulting in their phosphorylation, dimerization, and nuclear transport[2]; whereupon they show high-affinity DNA-binding[3]. Type I interferons signal through STAT1/STAT2 heterodimers to activate genes that contain IFN-sensitive response elements (ISRE) in their promoters[4]. Similarly, type II interferon signals through STAT1/STAT1 homodimers that recognize genes containing gamma-activated sequences (GAS) in their promoters[5]. There is considerable overlap between these two pathways as many IFN stimulated genes (ISGs) contain both ISRE and GAS sequences[4,5]. Once bound to cognate response genes, transcriptional co-activators CBP/p300 and BRG1 interact with STAT complexes[6–8] and recruit DNA polymerase II to initialize gene transcription[9]. However, the structural basis of STAT1 dimer recognition by transcriptional co-activators is still largely unknown.

As a widespread and successful apicomplexan parasite, *Toxoplasma gondii* can infect and survive in almost all warm-blooded hosts where it resides within its host cell in a protective compartment called the parasitophorous vacuole. Interferon signaling plays a central role in *Toxoplasma* infection and while type II interferon plays a dominant role[10,11], type I interferon is also implicated in the control of chronic infection[12]. Infection by *T. gondii* blocks STAT1 mediated transcription[13,14], despite not altering STAT1 phosphorylation, dimer formation, nuclear import or DNA recognition[15]. This block is mediated by the secreted effector TgIST (Toxoplasma inhibitor of STAT1-dependent transcription) that disrupts type I and type II interferon-mediated gene expression[12,16,17]. TgIST is exported from the parasitophorous vacuole and transported to the host nucleus where it interacts with STAT complexes bound to GAS sequences on the DNA. In addition, TgIST recruits a nucleosome remodeling and deacetylase complex known as Mi-2/NuRD, which is known for its role in repressing gene expression during development[18], suggesting that chromatin modification may contribute to altered expression[14,15]. A number of pathogens have been shown to disrupt STAT1 signaling[19,20] by a variety of different mechanisms. However, these prior findings do not address how TgIST, which binds to STAT1 complexes that are poised on chromatin at correct transcriptional start sites, is able to block transcription. Moreover, TgIST like many other pathogen secreted effectors, is an intrinsically disordered protein (IDP), complicating the analysis of its function. Such disordered proteins exhibit several features that facilitate their roles in transcriptional regulation and cell signaling including flexible conformation that allows for permissive partner interactions and the ability to function dynamically in regulatory networks[21]. Hence the study of pathogen IDPs may inform us about how such proteins evolve and function to modulate host signaling pathways.

In the present study, we investigate how the pathogen effector TgIST blocks STAT1-mediated gene expression. Using a combination of cellular, biochemical, and structural studies, we identify a repeat region in TgIST that mediates specific binding to the phosphorylated STAT1 dimer. We further define a core sequence in the TgIST repeat that is both necessary and sufficient to bind to STAT1 and block transcription. Biophysical and structural analyses reveal that the repeat region binds in a groove formed by two loops of the SH2 domain in STAT1, thus displacing the transcriptional co-activators CBP/p300. Our studies provide new insight into STAT1-mediated gene expression and define a unique mechanism for how pathogens selectively disrupt interferon signaling.

## Results

**TgIST recruits STAT1 and Mi-2/NuRD using different domains.** To dissect the interactions between TgIST and host proteins that underlie its inhibition of IFN-γ signaling, we performed immunoprecipitation (IP) experiments to determine if TgIST binds the Mi-2/NuRD complex independently of STAT1 or only as a ternary complex. Human sarcoma cell lines (U3A, STAT1 deficient; U3A-STAT1, STAT1 complemented) were infected with TgIST-Ty expressing parasites for 16 h and then activated with IFN-γ (100 U/mL) for 1 h. Nuclear extracts were used to capture TgIST-Ty by IP and co-precipitating proteins were analyzed by western blot. Consistent with a previous report[17], TgIST interacted with STAT1 in IFN-γ treated U3A-STAT1 cells and also co-immunoprecipitated components of the Mi-2/NuRD complex MTA1 and HDAC1 (Fig. 1a). The interaction between TgIST and STAT1 was dependent on IFN-γ, although this was not the case with binding to Mi-2/NuRD (Supplementary Fig. 1). TATA-binding protein (TBP), another component of nuclear extracts, did not interact with TgIST-Ty (Fig. 1a). TgIST-Ty also efficiently immunoprecipitated HDAC1 and MTA1 in U3A cells that lack STAT1 (Fig. 1a), indicating that TgIST binds to each complex separately.

To further explore host protein complexes interacting with TgIST, liquid chromatography-tandem mass spectrometry (LC-MS/MS) was performed on TgIST IP'd samples from cells described in Fig. 1a. We compared proteins IP'd by TgIST in U3A cells that do not express STAT *vs.* U3A-STAT1 expressing cells both in the absence and presence of IFN-γ. Uninfected but IFN-γ treated samples were used as a control to filter non-specific interactors. Proteins that were represented by a minimum of ≥2 peptides with 99% identify threshold from three independent TgIST IP replicates were analyzed to identify proteins that were significantly enriched in U3A-STAT1 expressing cells ($P \leq 0.05$ unpaired Student's $t$-test, two-tailed) (Supplementary Data 1, 2). Significantly enriched proteins were then subjected to protein-protein network analysis using the STRING database[22] to identify putative interactions within host proteins. Consistent with the IP and western blot results, STRING network analysis indicated a highly significant interaction of TgIST with HDAC1 and MTA1 and other Mi-2/NuRD components (Fig. 1b). The interaction between TgIST and Mi-2/NuRD was STAT1-independent since the identical Mi-2/NuRD complex was detected when STAT1-deficient U3A cells were used (Fig. 1c). We did not detect significant interactions with other transcription factors or chromatin-modifying complexes (Supplementary Data 1, 2).

**Identification of STAT1 and Mi-2/NuRD binding domains in TgIST.** TgIST is predicted to be an intrinsically disordered protein that lacks conserved folded domains or sequence similarity to other known proteins, thus complicating the analysis of function. TgIST contains an N-terminal hydrophobic sequence important for secretion and several nuclear localization sequences (NLS), a composition that is highly conserved among a variety of strains representing all the major lineages of *T. gondii*[23] (Supplementary Fig. 2a). Interestingly, several strains of *T. gondii* contain ~ 40 amino acid region that is repeated twice in tandem including the type I GT1 and RH strains, although type II strains ME49 and Pru contain only one copy of the repeat region. The arrangement and phylogenetic relatedness of the tandem repeats suggests that they arose by several independent duplications in different lineages (Supplementary Fig. 2b). The importance of the repeat structure is described further below, but even strains with a single sequence of this element are able to block IFN-γ signaling[17].

To explore the function of different regions of TgIST, we generated a series of C-terminal truncations of the type I allele of

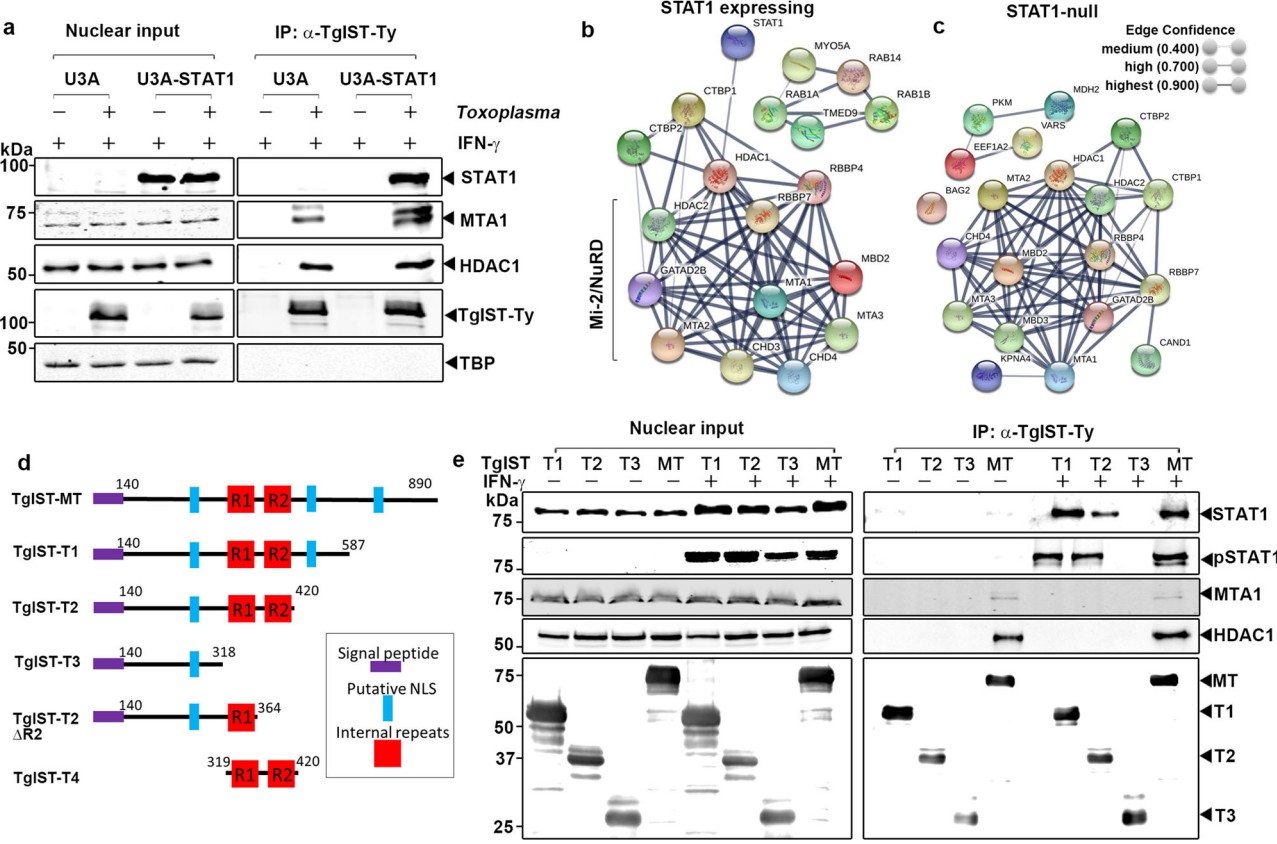

**Fig. 1 Different domains of TgIST interact with phosphorylated STAT1 dimers and the Mi-2/NuRD complex. a** Western blot analysis of host proteins following immunoprecipitation (IP) of TgIST-Ty from U3A (STAT1-null) or U3A-STAT1 (STAT1 complemented) cells that were infected with *T. gondii* expressing TgIST-Ty *vs.* mock for 16 h followed by treatment of 100 U/mL IFN-γ for 60 min. Two core components of the Mi-2/NuRD complex, metastasis-associated protein (MTA1) and histone deacetylase 1 (HDAC1), were co-precipitated with TgIST-Ty. TATA-binding protein (TBP) was used as a negative control. Representative blots of two independent experiments with similar results are shown here. **b, c** TgIST-associated host proteins identified by mass spectrometry analysis and visualized by STRING protein interaction network analysis. Immunoprecipitated TgIST-Ty from U3A-SAT1 cells (STAT1 expressing) and U3A cells (STAT1 null) were eluted from Dynabeads G and subjected to MS/MS analysis. STRING network derived from three experiments (≥2 peptide with 99% identity) including proteins with significant enrichment (*P* ≤ 0.05 unpaired Student's *t*-test, two-tailed). **d** Schematic representation of the mature form of TgIST (TgIST-MT) and truncated constructs (TgIST T1-T4) used in this study. The signal peptide (purple), putative nuclear localization sequences (NLS) (blue), and internal repeats R1 and R2 (red) are represented. **e** Western blot analysis of TgIST-Ty immunoprecipitation (IP) from TgIST transfected HEK293T cells. Cells were transfected with plasmids expressing different TgIST domains shown in **d** (i.e., TgIST-T1, -T2, -T3) or mature TgIST (TgIST-MT) for 23 h, then treated ± IFN-γ (100 U/mL) for additional 60 min prior to nuclear extract preparation. Membranes were incubated with corresponding primary antibodies as indicated and then IR dye-conjugated secondary antibodies. Visualization was performed using an Odyssey infrared imager. Representative blots of two independent experiments with similar results are shown here. Source data are provided in the Source Data file.

TgIST and expressed them transiently in HEK293T cells, and tested binding to STAT1 and Mi-2/NuRD. Full length and truncated constructs of TgIST were designed to initiate downstream of the TEXEL processing site that is cleaved by the ASP5 protease[24] during export in *T. gondii* (Fig. 1d). Sequential C-terminal deletions were created to either contain both central repeat domains (TgIST-T1 and TgIST-T2), lack the second repeat (TgIST-T2ΔR2), or lack both repeats (TgIST-T3). All four of these constructs contained at least one nuclear localization sequence (NLS). Truncated TgIST proteins containing a C-terminal Ty-tag were transfected into HEK293T cells and immunoprecipitated using an anti-Ty antibody. Total and phosphorylated STAT1 were readily co-precipitated with the mature form of TgIST (TgIST-MT) in IFN-γ treated nuclear factions but not in untreated cells, consistent with TgIST only binding to the phosphorylated STAT1 dimer that forms after INF-γ treatment (Fig. 1e). Furthermore, we found the interaction between STAT1 and TgIST relied on the repeat region based on the findings that truncations containing the repeats (i.e. TgIST-T1, TgIST-T2) pulled down pSTAT1 while

a construct lacking the repeats (i.e., TgIST-T3) did not (Fig. 1e). The core components of Mi-2/NuRD complex, MTA1 and HDAC1, co-immunoprecipitated only with the mature form of TgIST and this interaction was IFN-γ independent (Fig. 1e). Thus, these data indicate that the repeat region of TgIST mediates binding to STAT1 dimers, while independently, the C-terminal region is responsible for the recruitment of Mi-2/NuRD transcription repression complex.

**The repeat region of TgIST is sufficient to block IFN-γ signaling.** To examine the ability of TgIST to directly interact with STAT1, we co-expressed them as recombinant proteins in *E. coli*. We expressed the wild-type STAT1 core fragment that was previously used for crystallization[25], or a double cysteine mutated version (A656C and N658C, referred to as STAT1cc) that generates a locked dimer (STAT1cc) independent of phosphorylation[26]. STAT1 monomer, or STAT1cc dimer, were expressed as tag-free forms together with the different 6XHis-

tagged TgIST constructs shown in Fig. 1. Capture of His-tagged TgIST by nickel chromatography was used to assess STAT1 binding by SDS-PAGE and Coomassie blue staining. These studies revealed that TgIST requires the internal repeats for STAT1 binding as both TgIST-T1 and TgIST-T2 pulled down STAT1cc, while TgIST-T3 did not (Fig. 2a). Furthermore, the dimer form of STAT1cc was required for the interaction with TgIST as all three constructs failed to pull down STAT1 monomer (Fig. 2a). To further validate the necessity of the repeats in STAT1 binding, we compared the binding of a construct containing only the first repeat (TgIST-T2ΔR2) with TgIST-T2 that contains both repeats, as well as TgIST-T4 that contains both repeats but lacks the N-terminus, as shown in Fig. 1d. Both TgIST-T4 and TgIST-T2 co-purified STAT1cc much more efficiently than TgIST-T2ΔR2, suggesting that although one repeat is sufficient for binding, the two repeats act cooperatively (Fig. 2b). Collectively, these findings indicate that the internal repeats are both necessary and sufficient for binding to STAT1 and they only bind to the STAT1cc dimer form.

To examine the role of the repeats in blocking IFN-γ signaling, TgIST-GFP fusion constructs based on similar truncations to those described above were transfected into Hela cells treated with IFN-γ and the expression of IRF1 was visualized by immunofluorescence microscopy. Surprisingly, truncated forms of TgIST-T1 and TgIST-T2, as well as a truncated form of TgIST that harbors internal repeats but lacks the C-terminal Mi-2/NuRD binding domain, were still able to block IFN-γ signaling (Fig. 2c). However, TgIST-T3, which lacks the repeat region, did not block the expression of IRF1 despite its normal trafficking to the host nucleus (Fig. 2c). Deletion of one of the repeats in TgIST-T2-ΔR2 still led to a block in the expression of IRF1 (Fig. 2c), in agreement with its ability to bind STAT1cc (Fig. 2a). To evaluate the efficacy of IRF1 repression among different constructs, we monitored expression by quantitative imaging. There was no significant difference between the constructs containing repeats (i.e., TgIST-MT, TgIST-T1 and TgIST-T2), however, the IRF1 intensity of cells expressing TgIST-T3, which lacks repeats, was significantly elevated from the other constructs (Fig. 2d). As expected, the construct TgIST-T2ΔR2 was less effective in blocking IRF1 induction than constructs bearing two repeats (Fig. 2d), again suggesting the repeats act cooperatively. The differences in IRF1 induction were not due to differences in STAT1 expression levels that were similarly slightly decreased in cells expressing each of the constructs (Supplementary Fig. 3).

**Mapping the minimal STAT1 binding domain of TgIST.** To map the binding interface, purified TgIST-T2 complexed with STAT1cc was subjected to limited proteolysis using trypsin, followed by SDS-PAGE separation of resistant fragments (Supplementary Fig. 4). Six partial digestion products referred to as S1–S6 (Supplementary Fig. 4a) spanning from high to low molecular weight were isolated from the gel and subjected to mass spectrometry (MS) analysis. All of the limited proteolysis fragments contained a 7-amino acid region TALDV(F/L)R that is found at the core of both repeats (Supplementary Fig. 4b). To confirm the importance of the seven residue core region in the repeat, we changed all the residues to Ala in the construct that only contains the second of the two repeats (i.e., TgIST-R2) to generate the mutant called TgIST-R2-M1 (Fig. 2e). Compared to the wild-type TgIST-R2, mutated TgIST-R2-M1 lost the ability to bind to STAT1cc in the co-expression assay, confirming that this core region is necessary for STAT1 dimer binding (Fig. 2f). We also examined the role of the tandem repeats, and the 7-residue core, within TgIST in blocking IRF1 expression induced by IFN-γ using transient transfection in HeLa cells. TgIST-R2

which only contains the second repeat with an NLS sequence (NLS-R2) partially blocked IRF1 induction while TgIST-T4, which contains both repeats with the addition of a NLS (NLS-T4), was much more efficient in blocking IRF1 induction (Fig. 2g). To test whether the core 7 amino acids at the center of the repeat were necessary to block induction, we mutated these residues to Ala (similar to that shown in Fig. 2e) in both repeats of the TgIST-T2 construct to generate the mutant TgIST-T2-M2. This construct lost the ability to block IRF1 induction, demonstrating that repeats are necessary to block STAT1-mediated transcription (Fig. 2f, g). Taken together, these experiments define a core region within the repeats of TgIST that mediates binding to STAT1 and is both necessary and sufficient to block IRF1 induction.

**TgIST lacking Mi-2/NuRD binding still triggers repression of STAT1 signaling both in vitro and in vivo.** To further explore the immunological consequences of blocking STAT1 transcription by the different domains of TgIST, we complemented the deletion mutant with either full-length TgIST or TgIST-T1 (Fig. 1), which lacks the C-terminal Mi-2/NuRD binding domain. In complemented lines of both RH and Pru strain Δ*Tgist* knockouts, TgIST and TgIST-T1 were exported normally, trafficked to the host nuclei, and subsequently blocked expression of IRF1 (Supplementary Fig. 5). We then tested the influence of infection with the Δ*Tgist* mutant or complemented lines in the Type I RH strain on surface expression of major histocompatibility complex class I (MHC I) in macrophage RAW264.7 cells by flow cytometry (Fig. 3a, b and Supplementary Fig. 6). Activation with 100 U/ml IFN-γ increased the expression of I-A/I-E molecules on the surface of RAW264.7 cells. As expected, infection with wild-type parasites significantly reduced the upregulated expression of MHC I molecules (Fig. 3a, b). The downregulation of MHC I was TgIST dependent as shown by the recovery expression of MHC I when IFN-γ activated cells were infected with the Δ*Tgist* mutant (Fig. 3a, b). Interestingly, complementation with either wild-type TgIST or TgIST-T1 inhibited surface upregulation of MHC I in IFN-γ activated cells, regardless of the presence of the Mi-2/NuRD binding domain (Fig. 3a, b). These results indicate STAT1-binding by the N-terminal region of TgIST is sufficient to block primary (e.g., IRF1) and secondary (e.g., MHC I) responses that are stimulated by IFN-γ.

Since the N-terminal portion of TgIST is sufficient to inhibit IFN-γ signaling, we were curious if it would also be sufficient to restore virulence in vivo. We compared wild-type, mutant, and complemented lines in the type II Pru strain during infection of C57/BL6 mice, which provides a model for monitoring intermediate levels of virulence in vivo. The Δ*Tgist* mutant exhibited decreased virulence as shown by reduced parasite load and faster the recovery rates from infection when compared to infection with the wild-type strain (Fig. 3c, d). The decreased virulence of the Δ*Tgist* mutant was restored in parasites complemented the wild-type TgIST or the TgIST-T1 truncation (Fig. 3c, d). Although there was a slight delay in the kinetics of infection, the complemented strains reached the same peak burden compared to wild-type parasites (Fig. 3c, d). Hence, the restoration of the virulence by complementation with TgIST was also independent of Mi-2/NuRD binding.

**A core sequence in the repeats of TgIST mediated binding to the phosphorylated STAT1 dimer.** To facilitate further biochemical and structural studies, we expressed N-terminal Strep-tagged STAT1 in *E. coli* TKB1 cells that co-express ELK kinase to generate phosphorylated STAT1 dimers (pSTAT1d). Purified pSTATd was mixed with a double-stranded oligo of the GAS sequence and subjected to multiangle laser light scattering with

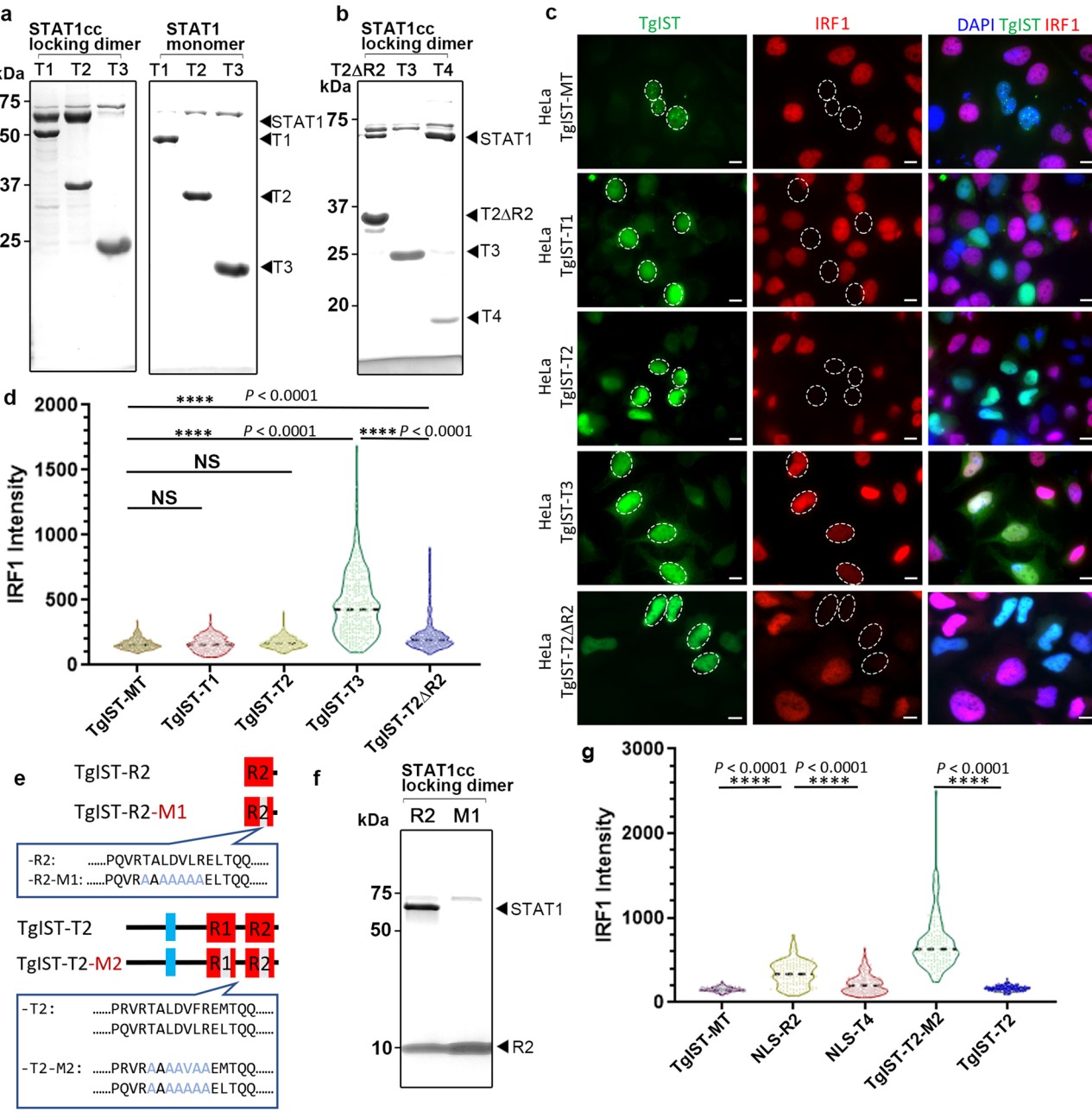

**Fig. 2 The TgIST repeats are required for STAT1 dimer binding and inhibition of IFN-γ signaling in host cells. a**, **b** Co-purification of STAT1cc or STAT1 with TgIST constructs as defined in Fig. 1d following co-expression in *E. coli*. Co-purifications were performed by capturing TgIST His₆-tagged constructs with nickel resin. SDS-PAGE gels stained with Coomassie blue. Representative gel images of two independent experiments with similar results are shown here. **c** Representative images showing induction of IRF1 in transfected Hela cells. Cells transiently expressing GFP-tagged TgIST constructs for 24 h were activated with IFN-γ for 6 h followed by staining for GFP (green), IRF1 (red) and DAPI (blue). Scale bar = 10 μm. Expression level of STAT1 remained unchanged (see Supplementary Fig. 3). Representative micrographs of two independent experiments with similar results are shown here. **d** Distribution of the IRF1 intensity collected from at least 250 TgIST-expressing HeLa. ****P ≤ 0.0001 with Kruskal–Wallis's test for multiple comparisons, and with Dunn's test for non-parametric correction. TgIST constructs correspond to those shown in (**c**) and are listed only with the suffix denting the construct. Data presented as violin plot and median (dotted line) from two independent experiments are shown. **e** Sequences of wild type TgIST-R2 containing a single repeat and a mutated version with an altered core region defined by changing TALDVLR to AAAAAAA (TgIST-R2-M1). Sequences of wild type TgIST-T2 containing two repeats and a mutated version with an altered core region defined by changing TALDVLR to AAAAAAA (TgIST-T2-M2). (**f**) TgIST-R2 and TgIST-R2-M1 tagged with His₆ were co-expressed with the STAT1cc locked dimer. Purification of the His-tagged TgIST proteins on nickel resin resulted in co-purification of STAT1cc with the wild-type TgIST-R2 (R2) construct but not the R2-M1 mutant (M1). SDS-PAGE gel stained with Coomassie blue. Representative gel image of two independent experiments with similar results are shown here. **g** The core repeat region was required for blocking IFN-γ signaling as detected by induction of IRF1. IRF1 intensity levels of HeLa cells stained as above. Data were collected from at least 250 TgIST-expressing (NLS-R2 or NLS-T4: R2 or T4 protein with nuclear localization sequence) HeLa cells and expressed as violin plots. ****P ≤ 0.0001 with Kruskal–Wallis's test for multiple comparisons, and with Dunn's test for non-parametric correction. Data presented as median (dotted line) from two independent experiments. Source data are provided in the Source Data file.

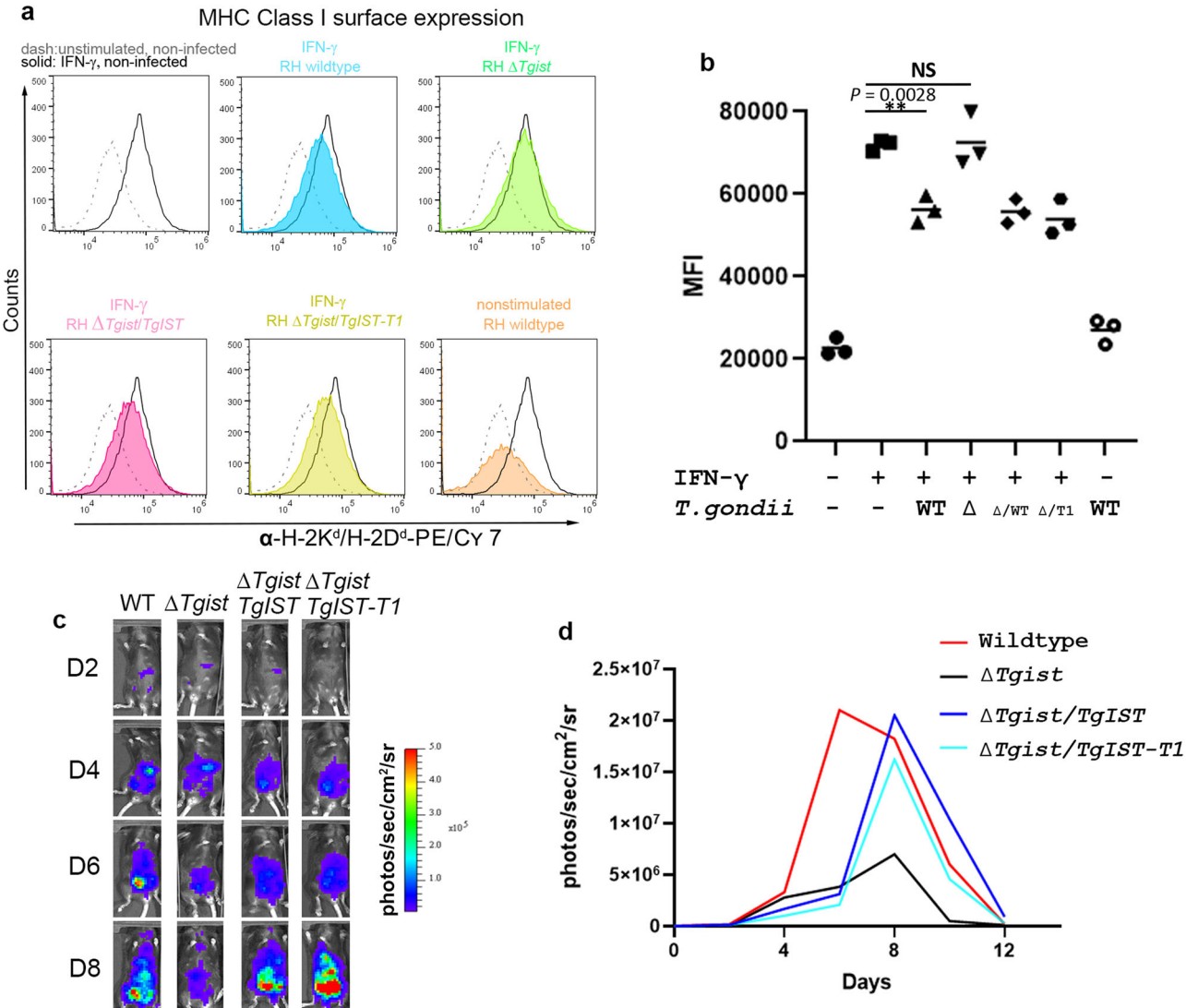

**Fig. 3 The TgIST repeats are sufficient to block IFN-γ signaling and promote parasites dissemination in vivo. a** Surface expression of MHC I on RAW264.7 macrophage cells by flow cytometry. RAW264.7 macrophages were infected with *T. gondii* for 6 h at a MOI = 3:1. Comparison of RH wild type, Δ*Tgist* mutant, and complemented lines expressing full length (RHΔ*Tgist/TgIST*) or a truncated version that binds STAT1 but not Mi2/NuRD (RHΔ*Tgist/TgIST-T1*). Expression of MHC I molecules I-A/I-E was measured on non-stimulated and IFN-γ stimulated RAW264.7 18 h after infection. Histograms show representative results of three independent experiments. **b** Median fluorescence intensities (MFI) were calculated from each of three independent experiments in (**a**). **$P = 0.0028$ with one-way ANOVA by Dunnett's test for multiple comparisons. **c** Comparison of C57/BL6 mice infected with firefly luciferase-expressing parasites. Comparison of Pru wild type, Δ*Tgist* mutant, and complemented lines expressing full length (Δ*Tgist/TgIST*) or truncated version that binds STAT1 but not NuRD (Δ*Tgist/TgIST-T1*). Bioluminescence was imaged with an in vivo imaging system from day 0 to 12 after the i.p. of 1000 tachyzoites. Representative mice from each group are shown at different days (D2, D4, D6, D8) post infection. Representative image of one mouse among five mice in each group are shown here. **d** Comparison of C57/BL6 mice infected with firefly luciferase-expressing wild-type and mutant parasites. The graph depicts mean whole-animal radiance from one experiment with five female mice in each group. The mouse with the lowest and highest intensity were excluded in each group. Source data are provided in the Source Data file.

in-line size exclusion chromatography (SEC-MALS) analysis to confirm the correct formation of the ternary complex (Supplementary Fig. 7 a–d). To validate the importance of the internal repeats of TgIST in binding to pSTAT1d, we monitored interactions using Bio-Layer Interferometry (BLI). We immobilized phosphorylated pSTAT1d or STAT1 monomer proteins on an Ni-NTA biosensor and monitored the interaction with purified TgIST-R2 or TgIST-R2-M1 in solution. pSTAT1d interacted strongly with TgIST-R2 as shown by the considerable increase of signal, while no binding was detected to STAT1 monomer (Fig. 4a). The mutated TgIST-R2-M1 construct showed dramatically decreased binding compared to the wild-type TgIST-R2

construct (Fig. 4a). These findings confirm that TgIST binds only to the phosphorylated STAT1 dimer, and that this interaction requires the 7 amino acid core shared by the repeats. To explore the stoichiometry of TgIST binding to pSTAT1d, we immobilized pSTAT1d on the Ni-NTA biosensor then immersed the loaded pins in purified TgIST-R2 peptide after GST removal. In this configuration, where monomeric TgIST-R2 in solution binds to dimeric STAT1 on the pin, the calculated affinity was $330 \pm 50$ nM using a 1:1 model for curve fitting (Fig. 4b). We also reversed the sample order and charged the biosensor with TgIST-R2 then interacted the pins with soluble pSTAT1d. In this configuration, where TgIST on the pin binds to dimeric pSTAT1d in

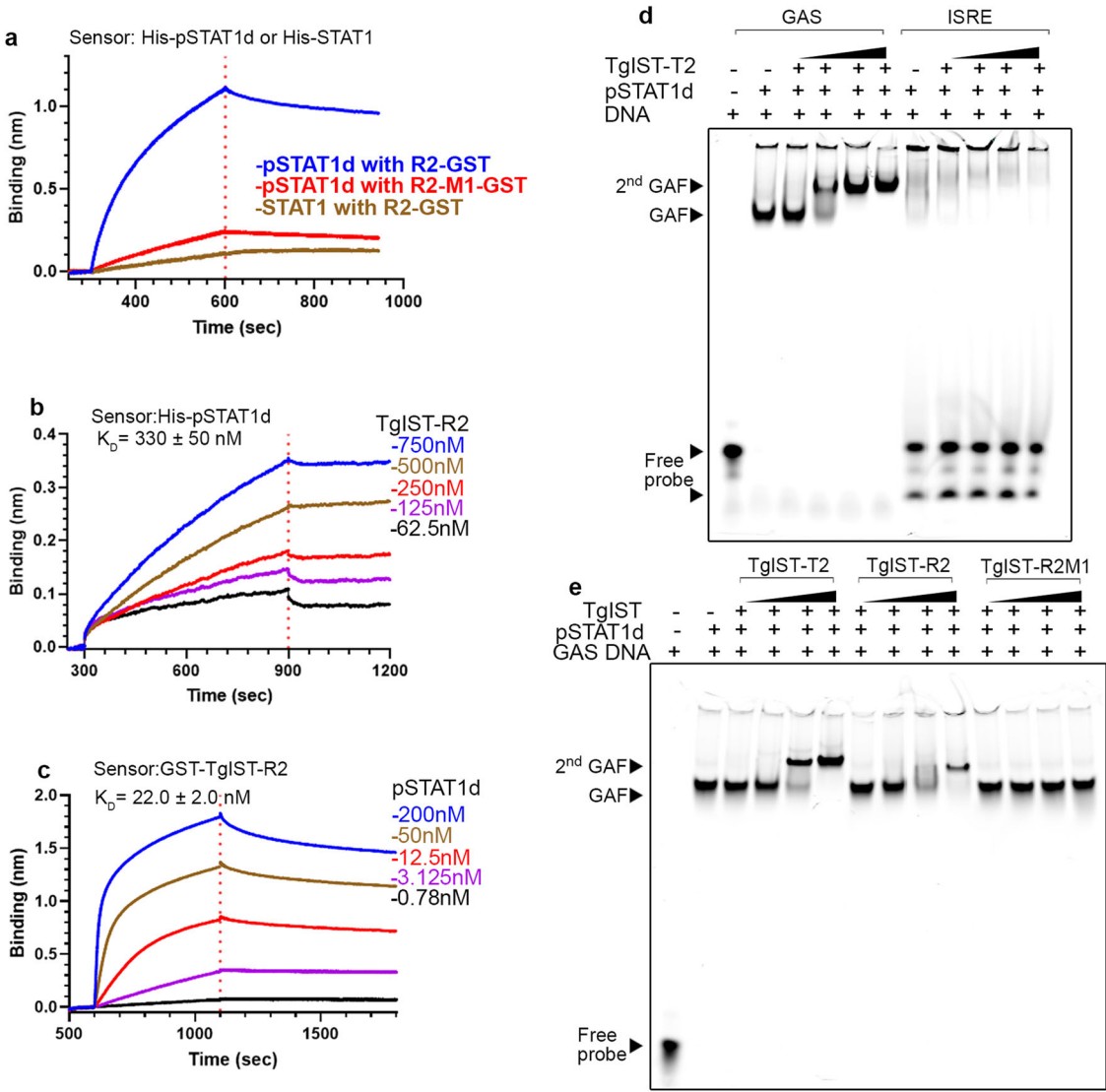

**Fig. 4 The repeat region of TgIST binds phosphorylated STAT1 dimers complexed with DNA. a** BLI sensorgrams obtained using biosensor loaded with His-tagged pSTAT1 dimer (pSTAT1d, 10 μg/mL) and incubated with 200 μM TgIST-R2 (R2-GST) or TgIST-R2-M1 (R2-GST-M1). The red dashed line separates the binding (left) and dissociation (right) phase. The binding of STAT1 monomer to TgIST-R2 (R2-GST) is shown in tan. **b** BLI sensorgrams using His-tagged recombinant phosphorylated STAT1 dimer (pSTAT1d) immobilized onto Ni-NTA biosensors, followed by incubation in various concentrations of TgIST-R2 after removal of the GST tag. Binding phase (left of the red dotted line) and dissociation phase (right of the red dotted line). **c** BLI sensorgrams using biosensor loaded with recombinant GST-tagged GST-TgIST-R2 and interacted with different concentrations of pSTAT1d. Binding phase (left of the dotted red line) and dissociation phase (right of the red dotted line). **d** Electromobility shift assays (EMSA) demonstrating TgIST-T2 binds to pSTAT1d associated with GAS DNA oligonucleotide but not to ISRE oligonucleotide. See also Supplementary Fig. 7 for unlabeled competitor assays. Purified pSTAT1 dimer was incubated with far-red fluorescence IRDye labeled double-stranded oligonucleotides, purified TgIST-T2 was then added, followed by a native polyacrylamide gel analysis. Visualization was performed using an Odyssey infrared imager. GAF refers to gamma-interferon activation factor composed of pSTAT1 and the labeled GAS DNA probe, 2nd GAF refers to GAF further complexed with TgIST-T2 resulting in a higher molecular weight complex. Black triangles indicate increasing concentration of TgIST-T2. **e** Formation of the 2nd GAF complex is dependent on the repeats in TgIST-R2. STAT1 dimer and GAS oligonucleotide complexes were made as in (**d**) but in addition to the T2 construct (TgIST-T2), GST-tagged TgIST-R2, or the mutated version TgIST-R2-M1, were also tested. GST was used as the affinity tag for purification and to increase the molecular weight in of TgIST-R2 to visualize R2-mediated super shift. See Supplementary Fig. 7 for GST alone control. Black triangles indicate increasing concentration of components added based on label at the top. Representative gel image (**d**, **e**) of two independent experiments with similar results were shown here. Source data are provided in the Source Data file.

solution, the apparent avidity was to $22.0 \pm 2.0$ nM when the curves were fit with a 1:2 model, consistent with cooperative binding by the STAT1 dimer (Fig. 4c).

We then examined the role of the TgIST repeats in interacting with pSTAT1d in complex with a GAS oligo to form the gamma activated factor (GAF), as detected by electrophoretic mobility shift assay (EMSA). The GAF complex readily formed when pSTAT1d was combined with a labeled GAS oligo, but not with

an ISRE oligo that normally binds to STAT1/STAT2 hetero-dimers (Fig. 4d). TgIST-T2 formed a super-shifted 2nd GAF complex in a concentration-dependent manner (Fig. 4d) that was competed by excess cold probe (Supplementary Fig. 7e). A similar super-shifted 2nd GAF was also observed when GST-tagged TgIST-R2 was used in the EMSA; however, the mutant TgIST-R2-M1 did not induce this complex (Fig. 3e, Supplementary Fig. 7f). These findings indicate that the interaction between

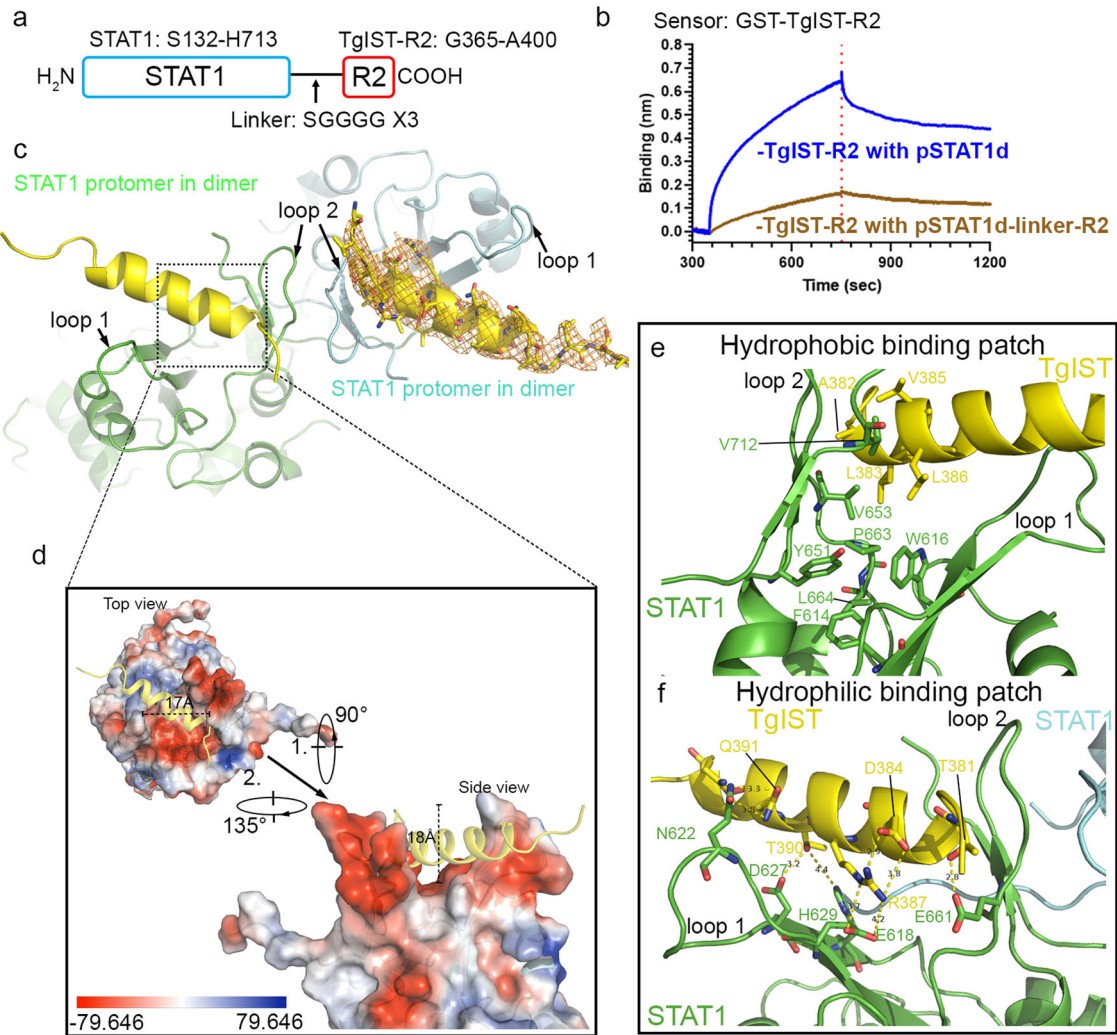

**Fig. 5 Molecular interactions between TgIST-R2 and STAT1 dimer revealed by structural studies. a** Schematic diagram of the STAT1-linker-R2 fusion protein (STAT1-TgIST-R2). The arrow indicates the flexible peptide linker. **b** BLI sensorgrams using biosensor loaded with recombinant GST-TgIST-R2 and interacted with pSTAT1d or pSTAT1d-linker-R2, both at 50 nM concentration. **c** Crystal structure of TgIST-R2 with STAT1 dimer based on the fused peptide construct shown in (**c**). The peptide backbones of two STAT1 protomers within the dimer are shown in cyan and green loops while the TgIST peptide is shown in yellow ribbon. Orange mesh represents the 2Fo-Fc map of TgIST-R2 contoured at 1.5 σ. **d** Electrostatic potential surface representation of the TgIST binding region in STAT1. **e, f** Direct interactions between TgIST-R2 and STAT1 at the binding interface. The yellow dashed lines indicate salt bridges or hydrogen bonds between corresponding atoms within 2.8–4.2 Å, with the exception of the 4.4 Å between TgIST-T390 and STAT1-H629. Source data are provided in the Source Data file.

TgIST and pSTAT1d bound to DNA requires the core 7 amino acids that are conserved in both repeats in TgIST.

**Crystal structure of TgIST-STAT1 represents a novel binding mode.** To reveal the mechanism of how TgIST recognizes pSTAT1d, we crystallized pSTAT1d complexed with DNA either alone or together with TgIST-R2 and obtained structures at 2.8 and 4.5 Å, respectively. A comparison of the structures revealed a new electron density located between two relatively unstructured loops of the SH2 domain of STAT1 (Supplementary Fig. 8a and 8b). Because this structure was of limited resolution, we fused TgIST-R2 to the C-terminus of STAT1. In crystals obtained from this construct, the peptide bound in the same place but was now visualized with much better resolution (Fig. 5a). This construct had a lower binding response to TgIST-R2 immobilized on the pin when compared to STAT1 alone, supporting the conclusion that the R2 peptide in this fused construct was bound in a functional state (Fig. 5b). Four X-ray diffraction datasets were

merged and scaled to yield a final model of the STAT1-TgIST-R2 structure at 2.9 Å resolution (Supplementary Table 1, Fig. 5c). TgIST-R2 adopted an α-helix and was bound in a groove between two loops formed by the SH2 domains of STAT1 (Fig. 5c). The density of the two loops, which are not clearly resolved in the structure of pSTAT1d alone and only partially seen in monomeric STAT1, are distinct in the new TgIST-R2 bound structure (Supplementary Fig. 8c). The binding of TgIST-T2 creates a 17 Å wide, 18 Å deep cavity on top of the protomer of the STAT1 dimer structure (Fig. 5d). Electrostatic potential analysis of the TgIST binding region on STAT1 reveals a negatively charged cavity at the bottom of two loops (Fig. 5d, red), in addition to a small hydrophobic patch. One STAT1 protomer shares a 551 Å² buried surface area with TgIST-R2, positioning the helix deep into the grove. After binding to TgIST-R2, the length of three β-strands (L601-F603, I612-W616, and H629-A630) in the SH2 domain of STAT1 was extended to form four additional hydrogen bonds (Supplementary Fig. 8d). Additionally, two small β-strands (W666-L667 and I671-D672) formed at the end of STAT1 that

were stabilized by three hydrogen bonds. These newly established hydrogen bonds alter and stabilize the conformation of loop 1 and loop 2 (Fig. 5e). In addition, hydrophobic amino acids in TgIST (i.e. L383, V385, and L386) are positioned opposite to the hydrophobic patch formed by F614, W616, Y651, V653, and P663 located at the bottom of loop 1, and L664 at the bottom of loop 2 in the STAT1 dimer (Fig. 5e). On the opposite side of the helix, TgIST interacts with STAT1 by means of multiple polar or electrostatic interactions. For example, R387 on TgIST forms a salt bridge with E618, and a hydrogen bond forms between T381 on TgIST with E661 on STAT1. In addition, residues D627 and H629 in STAT1 collectively form a hydrophilic surface on a small β-sheet, where D627 forms direct hydrogen bonds with T390 in TgIST (Fig. 5f). Finally, Q391 of TgIST, forms a hydrogen bond with N622 in loop 1 of STAT1, further strengthening the insertion of the helix into the cavity formed on top of STAT1.

**The repeat region of TgIST adopts a helical transition after binding to pSTAT1d.** Although full-length TgIST is predicted to be disordered, our crystal structure revealed that it adopts a helical conformation when bound to STAT1, a result consistent with secondary structure prediction of the repeat region (Supplementary Fig. 8e). To further explore the secondary structure of the repeat region, we examined purified TgIST-R2 by circular dichroism (CD). When suspended in buffered saline, the CD spectrum for TgIST-R2 suggests the protein is mainly unstructured, though the non-zero intensity between 210 and 240 nm indicates it is partially structured. When TgIST-R2 was suspended in buffered saline with increasing concentrations (0–40%) of trifluoroethanol (TFE), the CD signal increased, particularly between 205 and 225 nm, indicating a propensity to form an α-helix (Fig. 6a). Further analysis of the TgIST-R2 region using HeliQuest[27] indicated that the ten residues from V379 to E388 can form an amphipathic α-helix (Fig. 6b). We generated a mutant of TgIST-R2 that interchanged two amino acid pairs (VR

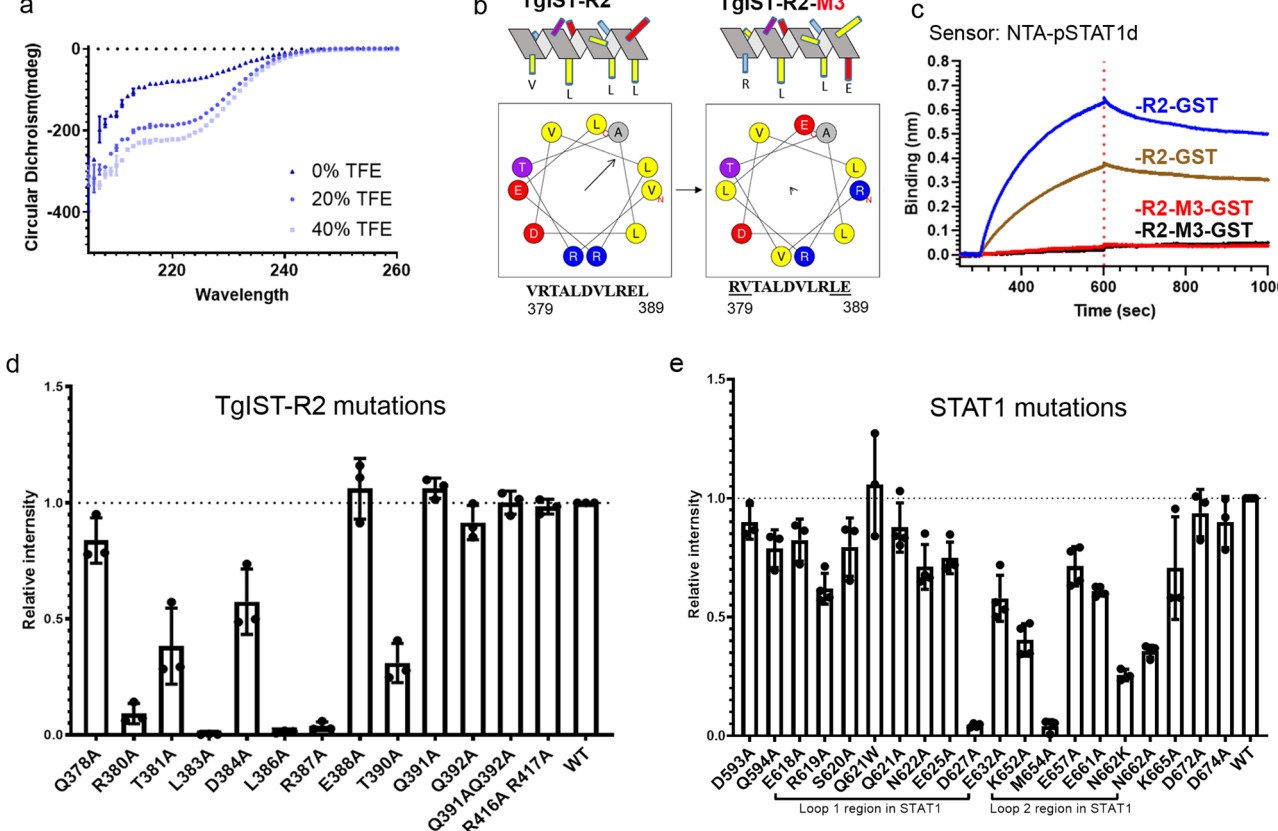

**Fig. 6 TgIST adopts a helix conformation upon binding the STAT1 dimer. a** Circular dichroism (CD) of TgIST-R2. Nickel affinity-purified TgIST-T2 was buffer exchanged into phosphate buffer (pH 7.8) and adjusted to a concentration of 100 μg/ml. CD spectra were recorded in a 10 mm cuvette at 25 °C in the absence of trifluoroethanol (TFE), or in the presence of 20%, 40% TFE that induced secondary structure formation. Data presented as mean + s.d. from two independent experiments. **b** TgIST-R2 is predicted to form an amphipathic alpha-helical structure shown as a side projection at the top and a helical wheel at the bottom. The wild type repeat region is shown on the left and the mutant is on the right. Predictions are based on the HeliQuest server. Arrows within the helical wheel projection indicate the hydrophobic face and the length is proportional to the degree of hydrophobicity. **c** Measurement of the interaction between wild-type TgIST-R2 (R2-GST) and pSTAT1d vs. the charge altered TgIST-R2 mutant (R2-M3-GST) and pSTAT1d by BLI assay. Phosphorylated STAT1 dimer (pSTAT1d) was immobilized onto Ni-NTA biosensors, followed by incubating with R2-GST (100 or 50 nM, represented by blue and tan, respectively) or R2-M3-GST (100 or 50 nM, represented by red and black, respectively). **d** Nickel affinity purification of His-tagged TgIST-R2 wild type (WT) and various point mutants of TgIST-R2 was used to test copurification with the STAT1 locked dimer (STAT1cc). Eluted fractions were separated by SDS-PAGE gel then stained with Coomassie blue (see Supplementary Fig. 9a). The relative binding intensities were adjusted to the binding between wild-type TgIST-R2 and STAT1cc. Data presented as mean + s.d. from three independent experiments (n = 3). **e** Nickel affinity purification of His-tagged TgIST-R2 and copurification of wild type (WT) and various mutants of STAT1 locked dimer (STAT1cc). Polar and charged amino acids in and around two loop regions were mutated in STAT1cc. Quantification was performed as in (**d**). Data presented as mean ± s.d. from three independent experiments (n = 3). Source data are provided in the Source Data file.

to RV at position 379–380, and EL to LE at position 388–389) to destroy the amphipathic nature of the peptide without altering the helical conformation (Fig. 6b). BLI assays confirmed that wild-type TgIST-R2 bound strongly to pSTAT1d, while the altered TgIST-R2-M3 peptide showed no binding (Fig. 6c). These data suggest that both the amphipathic nature of the repeat region helix and the placement of specific charged and hydrophobic residues are important for STAT1 binding.

To test the sidechain interactions observed in the co-crystal structure, mutations were introduced into a co-expression plasmid harboring 6XHis-tagged TgIST-R2 and STAT1cc, followed by purification of TgIST by nickel chromatography and detection of STAT1 binding by SDS-PAGE and Coomassie blue staining. Mutation of TgIST residues R380A and T390A significantly reduced binding to STAT1cc (93% and 99%, respectively), suggesting that they serve as N- and C-terminal anchors for the interaction (Fig. 6d and Supplementary Fig. 9a). Mutation of hydrophobic residues L383A and L386A in TgIST-R2 completely ablated binding to STAT1cc, suggesting that both the length and hydrophobic nature of the sidechain is important for the interactions. In addition, polar and charged residues such as T381, D384, R387 lost binding to STAT1cc to various degrees (from 43 to 99%, Fig. 6d), indicating the amphipathic property of the α-helix contributes to STAT1 binding. In summary, point mutations within the core binding domain region of TgIST-R2 support an α-helical amphipathic structure, and demonstrate the importance of key residues in this structure in binding to the STAT1 dimer.

We also explored mutations in STAT1 to define the binding interface. Mutations on STAT1 of D627A in loop 1 and M654A in loop 2 of STAT1 almost completely abrogated binding to TgIST-R2 (Fig. 6e and Supplementary Fig. 9b). The D627A mutation removes its interaction with T390 of TgIST, and might destabilize the composition of the β-sheet that maintains the charged binding grove. The M654A mutation is critical in forming the interface between two STAT1 protomers in the dimeric STAT1. In addition, mutations of K652 and N662 in loop 2 also reduced binding, while other mutations outside the two loops, had no effect on binding (Fig. 6e). The sidechain interaction between TgIST R387 and STAT1 E661 is further supported by the evidence that the STAT1 E661A mutation had partial binding defects to TgIST-T2. In summary, results from the point mutations corroborate observations from the co-crystal structures and suggest that TgIST-R2 binds STAT1 at its symmetrical dimer interface.

**The basis of STAT1 dimer recognition by TgIST**. To explain the mechanism of specific STAT1 dimer binding by the repeat region of TgIST, we superimposed the structure of the SH2 domain of the pSTAT1d dimer obtained here with the corresponding structures of monomeric STAT1 (PDB: 1YVL)[28] using sequence-based alignment. Interestingly, loop 2 is clearly present in monomeric STAT1, yet it shows a strikingly different orientation (Fig. 7a, orange colored close state) compared to the structure of protomers within dimeric STAT1 (Fig. 7a, cyan colored open state). These alterations are stabilized by extended β-sheets forming additional hydrogen bonds at the base of each loop (Supplementary Fig. 8d), as discussed above. In the free monomeric STAT1 structure, loop 2 is in a closed conformation, folding over towards loop 1, while it flips to an open conformation in the dimeric structure (Fig. 7a). Importantly, the position of loop 2 in the free monomeric structure partially occupies the proposed TgIST-T2 binding site, blocking entry into the grove formed by two loops in the pSTAT1 dimer (Fig. 7 a, b). Hence, the altered conformation of loop 2 in the dimeric

STAT1 structure exposes a surface for TgIST-R2 binding, that is otherwise absent in the STAT1 free monomer, thus providing an explanation for the specificity of TgIST-R2 in binding to the dimeric form of STAT1. The loop regions of STAT1 are highly conserved across a number of mammalian species and they are predicted to adopt a similar topology (Supplementary Fig. 10a, b) consistent with the ability of *T. gondii* to block IFN-γ induced gene expression from mouse to human[13–17].

Based on the importance of the SH2 domain of STAT1 in mediating interaction with TgIST, we made an alignment of other SH2 harboring proteins including STAT1, STAT3, and STAT6. The core composition of SH2 domain, which is composed of a large β-sheet flanked by two α-helices aligned well for all structures (Fig. 7c, d). STAT3 has a similar loop 2 orientation compared to STAT1, however, loop 1 is folded inward and it does not create a similar open groove seen in the STAT1 dimer. STAT6 lacks the loop 2 region that is found in STAT1 and STAT3 (Fig. 7d), suggesting sequence differences between STATs proteins affect the folding of this region. These structural variations help explain the specificity of TgIST for binding to STAT1.

**TgIST competes with the interaction between STAT1 and CBP/p300 coactivators**. The histone acetyltransferase coactivators CBP and p300 directly interact with STAT1 within its C-terminal transcriptional activation domain (TAD) domain to facilitate efficient transcription activation[29]. The TAD domain is located C-terminally to the STAT1 dimer interface (our structure and PDB: 1BF5[25]), which suggests that the binding of TgIST in the grove formed by the STAT1 dimer may be responsible for preventing the recruitment of CBP/p300. We tested this hypothesis by expressing constructs of TgIST in HEK293T cells and determining whether they could displace CBP/p300 from immunoprecipitated STAT1. We compared a form of TgIST containing two copies of the STAT1-binding repeat (TgIST-T2) to a similar copy where the core residues were mutated to alanine (TgIST-T2-M2) and a truncated construct that lacks the repeats (TgIST-T3). Although the efficiency of immunoprecipitation of STAT1 was the same, CBP/p300 was less efficiently coprecipitated in cells expressing TgIST-T2 that binds STAT1, when compared to the mutated TgIST constructs that do not bind STAT1 (Fig. 7e, f). A reciprocal experiment performed by immunoprecipitating Ty-tagged TgIST from HEK293T cell lysates confirmed that only TgIST-T2, and the mature form TgIST-MT, were able to interact with STAT1 (Supplementary Fig. 10c). Consistent with this result, label-free quantitative mass spectrometry showed less CBP/p300 co-precipitated with STAT1 in TgIST-T2 vs. TgIST-T2-M2 transfected cells (Supplementary Fig. 10d, e). Taken together, these results indicate that the repeat regions of TgIST bind to STAT1 and block the recruitment of CBP/p300.

**Discussion**
The intrinsically disordered protein TgIST is responsible for blocking both type I and type II interferon signaling by *T. gondii*, thus enhancing survival during acute and chronic infection[12,16,17]. TgIST binds both to phosphorylated STAT1 homodimers and STAT1/STAT2 heterodimers and recruits the chromatin repressive complex Mi-2-NuRD[12,16,17]. Our studies define two distinct regions of TgIST that function independently: a N-terminal region containing repeats bind directly to STAT1 dimers and a separate C-terminal region binds Mi-2/NuRD. The N-terminal repeat region was both necessary and sufficient to bind STAT1 dimers and to block transcription of IRF1 by IFN-γ. Cellular, biochemical, and structural studies reveal that the TgIST repeats adopt a helical conformation when bound to the STAT1

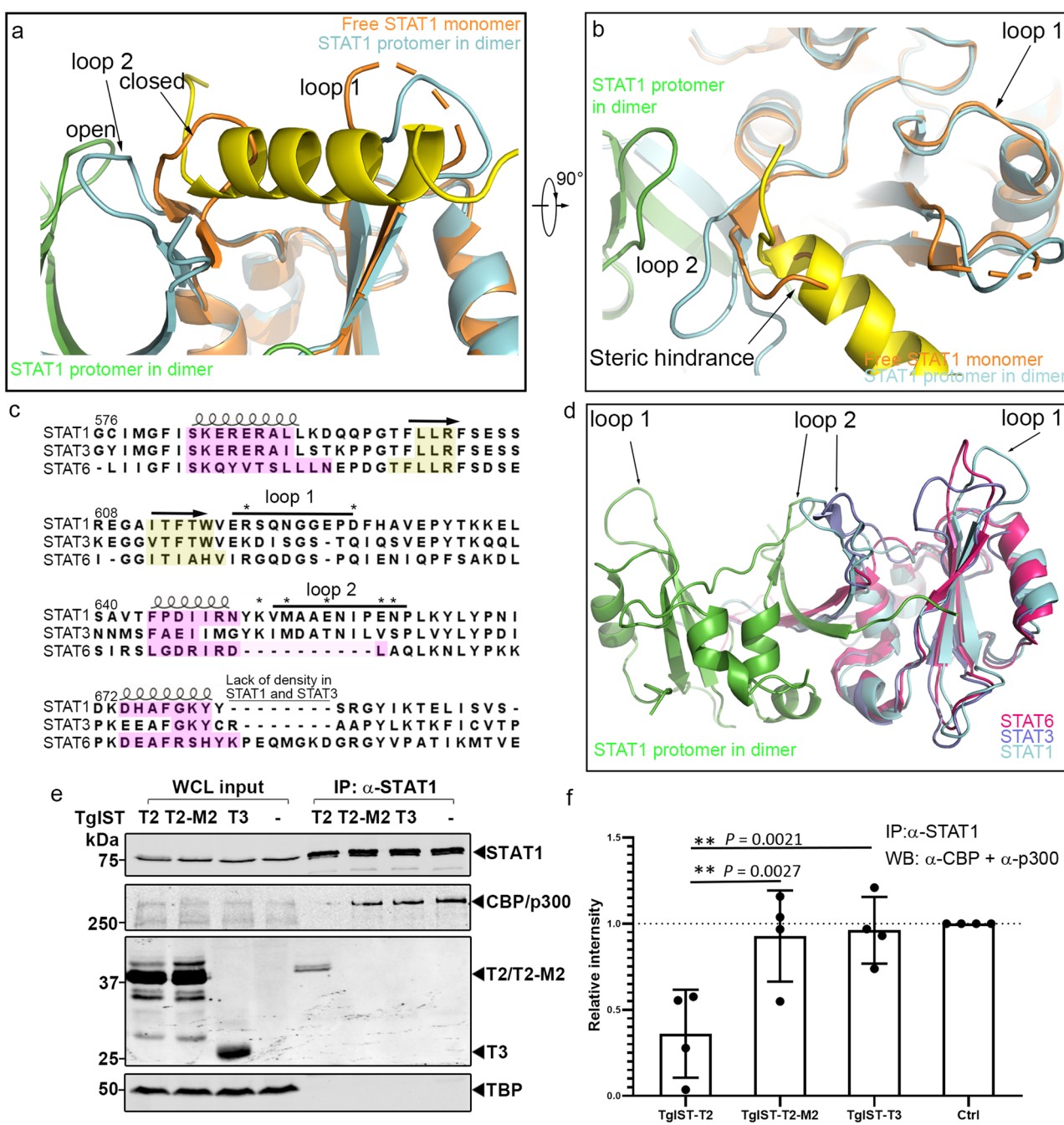

**Fig. 7 TgIST binding to the STAT1 dimer interface blocks recruitment of co-transcription factors CBP/p300. a, b** Structural superimposition of pSTAT1-TgIST-R2 complex and the free monomeric STAT1 (orange, PDB: 1YVL) showing steric hindrance at the SH2 domain. TgIST-R2-bound structure of pSTAT1 is shown for comparison with loop 2 in the open conformation (ycan). These features are conserved across species (see Supplementary Fig. 10). **c** Structure-based sequence alignment of the SH2 domains of phosphorylated STAT1, STAT3, and STAT6 dimers. Secondary structures are indicated using loops or arrows for α-helix and β-sheet, respectively. The yellow or magenta highlighted amino acids represent conserved secondary structures defined by the DALI server. Asterisks indicate residues that lost binding to TgIST-T2 when mutated in STAT1. **d** Structural superimposition of STAT1 (cyan, green), STAT3 dimer (purple, PDB: 1BG1), and STAT6 dimer (red, PDB: 5D39) highlighting differences in loop 2. **e** Western blot analysis of STAT1 immunoprecipitation (IP) from TgIST transfected HEK293T cells. Cells were transfected with plasmids expressing different TgIST domains: TgIST-T2 (T2) containing two STAT1-binding repeats; TgIST-T2-M2 (M2) containing mutated STAT1-binding repeats; TgIST-T3 (T3) lacking the two repeats (see Figs. 1d and 2e). Cells were grown for 23 h, then treated ±IFN-γ (100 U/mL) for an additional 60 min prior to whole-cell extract preparation. Membranes were incubated with corresponding primary antibodies as indicated and then IR dye-conjugated secondary antibodies. Visualization was performed using an Odyssey infrared imager. **f** Quantification and statistics of the CBP/p300 band intensities in (**e**). Intensities of the bands corresponding to CBP/p300 were measured by Image Studio then relative intensity was adjusted to CBP/p300 intensity in the mock transfect lane. Shapiro-Wilk test for normality; **$P$ = exact values shown as determined with two-way ANOVA by Dunnett's test for multiple comparisons. Data presented as mean ± s.d. from four independent experiments. Source data are provided in the Source Data file.

dimer interface. A change in the conformation of two loops in the SH2 domain of STAT1 during dimer formation exposes a new interface for binding of TgIST. In turn, TgIST binding alters the confirmation of STAT1 and prevents recruitment of transcription co-activator CBP/p300, thus revealing the molecular details of how the parasite effector TgIST specifically prevents STAT1-mediated transcription.

Although previous studies have shown that TgIST binds both to STAT1 and to Mi-2/NuRD, the relationship between these two interactions in blocking IFN signaling is uncertain, especially as changes in host chromatin modification differ between primary and secondary response genes[14,15]. Here we compared IPs of TgIST from STAT1 null to STAT1 expressing cells and found that TgIST interacts with the Mi-2/NuRD complex in the absence of STAT1 and without IFN-γ stimulation. Furthermore, we demonstrate using truncations that the C-terminus of TgIST contains the Mi-2/NuRD interacting domain, and that this region is dispensable for the ability of the N-terminal repeat-containing region to block IFN-γ induction of the primary response gene IRF1 as well as upregulation of MHC I. Although the recruitment of Mi-2/NuRD was not required for the block mediated by the N-terminal repeats, it may function in the regulation of other IFN-γ induced genes. Consistent with this model, it has recently been suggested that recruitment of Mi-2/NuRD to IFN-γ responsive genes is more important in secondary response genes where chromatin modulation may play a more important role[30].

In addition to TgIST, a number of other *T. gondii* effectors released beyond the parasitophorous vacuole are intrinsically disordered proteins (IDPs) and they bind a range of different host molecules to disrupt signaling and gene expression[31,32]. The binding mechanism of most such IDPs to their cognate host targets is largely unknown. One exception is GRA24, which contains a well recognizable kinase interaction motif (KIM) within its two internal repeats that naturally adopt a helical structure when it was crystallized with its host target p38α MAP kinase[33,34]. In contrast, TgIST lacks any secondary structure when unbound and its repeat region only adopted a helical conformation on binding to the STAT1 dimer, a property of induced-fit that is similar to other IDPs[21]. A single repeat region of TgIST was sufficient to block STAT1, consistent with previous reports[17], while the duplicated repeat was more effective. Internal repeats are common in intrinsically disordered proteins where they likely arise by duplication, allowing rapid evolution of new functionality[35]. The flexible features of IDPs[21] appear to be advantageous for secreted virulence factors due to their ease of export across membranes, adaptable binding to different host targets, and ability to rapidly evolve in the absence of structural constraint.

The crystal structure of the repeat region of TgIST with pSTAT1d revealed the R2 peptide bound in a groove that forms at the dimer interface. The binding of the helical peptide of R2 within the STAT1 groove was stabilized by polar and hydrophobic interactions, as supported by the mutational analysis in both partners. We speculate that the full-length TgIST binds to this interface in a symmetrical orientation, since repeat 1 and repeat 2 share the same sequence in their core. The increased binding avidity measured by BLI also supports this model of cooperative binding. The groove occupied by TgIST-R2 sits on the top of the dimer interface formed by two loops in the SH2 domains of protomers in the dimeric STAT1 structure. Our findings suggest that during dimer formation, loop 2 undergoes a reorientation from an inwardly folded conformation in the free monomer, to an outward conformation in the dimer. The altered conformation requires tyrosine phosphorylation in each of the two STAT1 monomers, reorienting loop 2 regions from adjacent

symmetric monomers to stabilize the dimer structure through the interaction of their SH2 domains. The interaction between two SH2 domains exposes two symmetrical binding sites on one STAT1 dimer, each of which is likely occupied by one helical peptide of TgIST-R2. The obvious differences in the loop 2 regions among STAT1, STAT3, and STAT6 dimers likely determine selectively of TgIST, which blocks STAT1 function, but does not play a role in mediating signaling by STAT3 or STAT6 based on transcriptional studies[12,14,16].

CBP and p300 are very closely related paralogs that participate in the recruitment of transcription factors to regulate multiple cellular processes including inflammatory and immune responses[9]. CBP/p300 are scaffold proteins that contain extensive regions of disorder, interspersed with globular domains including TAZ1, KIX, and TAZ2, which interact with transactivation domains of various transcription factors[36]. The transcriptional activating domain (TAD) of STAT1 preferentially interacts with the TAZ2 domain of CBP/p300 to enhance activity and the structure of this complex has been determined by NMR[8]. Although it has not yet been possible to obtain a crystal structure of the TAD domain connected with STAT1, it lies at the C-terminus downstream of the SH2 domain in a position that would orient it above the dimer interface. Our findings indicate that the altered conformation of the SH2 domains of STAT1 bound to TgIST results in failure to recruit CBP/p300, likely by preventing interaction between the TAD and CBP/p300, which is normally required to recruit polymerase II and drive gene expression[37]. Hence, defining the molecular mechanism for the inhibition of STAT1-mediated transcription by TgIST provides new insight into the conformational-dependent recruitment of transcriptional co-activators in controlling gene transcription.

## Methods

**Animal studies and ethical approval**. Animal studies were conducted according to the U.S. Public Health Service policy on human care and the use of laboratory animals. Animals were maintained in a specific pathogen-free barrier facility approved by the Association for Assessment and Accreditation of Laboratory Animal Care. Animal studies were approved by the Institutional Care Committee, Division of Comparative Medicine, Washington University. C57BL/6 (strain 000664) mice were purchased from Jackson Laboratory and bred locally at Washington University. Control and infected animals were housed in separate isolator cages in the same facility. Female mice between 8 and 10 weeks of age were used to perform experiments. Groups of 5 C57BL/6 mice per experimental group were infected i.p. with 1000 firefly luciferase-expressing parasites, imaged days 0–12 using a SpectrumBL in vivo optical imaging system (Perkin Elmer) and analyzed with the Living Image software (Perkin Elmer). Mice were anesthetized with 2% isoflurane and injected i.p. with D-luciferin (150 mg/kg) prior to imaging. At the end of the experiment, animals were euthanized with carbon dioxide using a SMARTBOX auto CO2 unit as approved by the Association for Assessment and Accreditation of Laboratory Animal Care.

**Bacterial cultures**. *Escherichia coli* strain TKB1 (Agilent) was used to express the phosphorylated STAT1 dimer. *E. coli* strain Rosetta (DE3) (Novagen) was used to produce recombinant proteins. Expression plasmids and cloning primers used in this study are listed in Supplementary Data 3, 4. The sequence of TgIST from the type I strain GT1 was used for constructing plasmids used for expression in *E. coli*, mammalian cells, or different strains of *T. gondii*.

**Parasite and host cell cultures**. *T. gondii* parasites were propagated in HFF (human foreskin fibroblasts) cells as described previously[17]. Plasmids expressing TgIST or TgIST-T1 were co-transfected with a CRISRP/Cas9 sgRNA plasmid targeting the UPRT locus (Supplementary Data 3, 4) into GFP-expressing Type I RHΔ*Tgist*, or luciferase-expressing Type II PruΔ*Tgist* parasites[17] and stable transgenic parasites were selected in pyrimethamine (3 mM). HFF, HeLa and 293T cells were maintained in Dulbecco's modified Eagle's medium (Invitrogen) supplemented with 10% HyClone fetal bovine serum (GE Healthcare Life Sciences), 10 μg/mL gentamicin (Thermo Fisher Scientific) and 10 mM glutamine (Thermo Fisher Scientific). Cultures were tested for *Mycoplasma* contamination using the e-Myco plus mycoplasma PCR detection kit following the manufacturer's manual (Boca Scientific). Transient transfections for protein expression in 293T were performed with Transit-LT1 transfection reagent (Mirus Biotech) according to the manufacturer's instructions.

**Immunoprecipitation and western blotting**. The type I RH ΔTgist mutant expressing epitope-tagged TgIST-Ty was used in the immunoprecipitation experiments (MOI = 3:1). For immunoprecipitation of Ty-tagged proteins, nuclear extracts were prepared using the NE-PER kit (Thermo Fisher Scientific), incubated with anti-Ty mAb BB2 (5 μl antibody per 200 μl nuclear extracts)[38] at 4 °C for 2 h, then incubated with ProteinG Dynabeads (Thermo Fisher Scientific) overnight at 4 °C. For immunoprecipitation of STAT1, whole-cell lysates were prepared using Pierce™ IP Lysis Buffer (Thermo Fisher Scientific), incubated with anti-STAT1 mAb (5 μl antibody per 500 μl whole cell extracts) at 4 °C for 2 h, then incubated with ProteinG Dynabeads (Thermo Fisher Scientific) for additional 1 h at 4 °C. Samples were eluted using 50 mM glycine (pH 2.8) after three times washes with PBS. The input extracts (5% of total) and immunoprecipitated samples (10% of total) were separated using 8–12% acrylamide gels and transferred onto a nitro-cellulose membrane for western blotting. The membrane was blocked using 5% milk diluted in PBST (Phosphate buffered-saline with 0.05%v/v Tween-20) and probed with primary antibodies for overnight at 4 °C followed by three washes with PBST.

**MS/MS analysis**. Dynabeads from IP experiments were suspended in ammonium bicarbonate, then reduced with 2 mM DTT at 37 °C for 1 h. Subsequently, alkylation was performed in 10 mM iodoacetamide for 20 min at 22 °C in the dark. Sample digestion by trypsin was carried out overnight at 37 °C followed by drying and redissolving in 2.5% acetonitrile and 0.1% formic acid. Samples were analyzed by nanoLC-MS/MS using a 2 h gradient on a 0.075 mm × 250 mm C18 Waters CSH column feeding into a Q-Exactive HF mass spectrometer. Data were analyzed by Mascot (Matrix Science) to search SwissProt Homo sapiens and ToxoDB-28_TgondiiME49_Annotated Proteins using a fragment ion mass tolerance of 0.060 Da and a parent ion tolerance of 10.0 PPM. Deamidated of asparagine and glutamine and oxidation of methionine were specified in Mascot as variable modifications. Scaffold software (Proteome Software Inc.) was used to validate and visualize MS/MS results and to define protein matches.

**Sequence analysis, alignment, and graphical presentation**. Nuclear localization sequences (NLS) were identified in TgIST based on cNLS mapper[39]. Sequences of STAT1 were retrieved from the Uniprot database for human, P42224; mouse, P42225; rat, Q9QXK0; pig, Q764M5; gorilla, G3SFV1; bovine, A0A3Q1ME65; cat, A0A337SVH3; chicken, Q5ZJK3; chimpanzee, A0A2I3TNY5; horse, A0A3Q2L3I5; and dog, A0A5F4C2J9. For multiple sequence alignment, STAT1 protein sequences imported into Jalview software[40] and the multiple sequence alignment was performed by Muscle[41] using default settings. Homology modeling of STAT1 from different species were performed by Swiss-model server using STAT1 dimer structure (PDB: 1BF5) as a template. All models were subsequently aligned and visualized by Pymol[42].

The SH2 domain of STAT1 (PDB: 1BF5), STAT3 (PDB: 1BG1) and STAT6 (PDB: 5D39) were obtained from their corresponding PDB files and imported into the DALI server[43] for structural comparison. Amino acids represent conserved secondary structures defined by the DALI server were manually highlighted.

**Immunofluorescence microscopy**. HeLa cells were grown on glass coverslips (for qualitative assays) or 96-well optical clear plates (Greiner) (for automated quantitative analysis). For conventional microscopy, transfected cells were fixed with 4% formaldehyde in PBS, permeabilized with 0.1% Triton X-100, blocked with 5% fetal bovine serum and 5% normal goat serum, labeled with primary antibodies followed by Alexa fluor-conjugated secondary antibodies (1:1000 dilution), and mounted with Prolong Gold with DAPI. Images were captured and analyzed with a Zeiss Axioskop 2 Plus wide-field fluorescence microscope and ZEN software (Carl Zeiss, Inc.). For automated quantitative assays, cells were stained with primary antibodies followed by Alexa fluor-conjugated secondary (1:1000 dilution) antibodies and finally stained with DAPI. Full details of antibodies and dilutions are listed in Supplementary Table 2. Images were captured using a Cytation 3 multimode plate imager with Gen5 software (BioTek). Following the capture of fluorescence signals using the Cytation 3, the intensity of IRF1 was analyzed using CellProfiler software[44]. In order to measure the IRF1 intensity only in TgIST transfected Hela cells, the GFP channel was analyzed to identify transfected cells, followed by measurement of IRF1 in regions defined by DAPI-positive nuclei.

**Protein expression and purification of TgIST and STAT1**. A portion of the STAT protein corresponding to residues S132-H713[25] and a lock dimer mutant containing two mutations at A656C and N658C (designated as STAT1cc)[26] were expressed in E. coli as recombinant proteins. A modified the pET-15b plasmid that encodes both TgIST and STAT1 with separate T7 promoters and terminators (also see Supplementary Fig. 3a for schematic presentation) was transformed in E. coli Rosetta (DE3) (Novagen) and cultured in Luria Broth Media. Co-expressed TgIST and STAT1 were induced by culture of E. coli Rosetta (DE3) with 0.1 mM IPTG for 15 h at 16 °C. Alternatively, TgIST constructs were produced as GST fusions in pET-15b propagated in E. coli Rosetta (DE3) induced with 0.1 mM IPTG and grown for 5 h at 30 °C. Cells were pelleted and lysed using Cellytic B (Millipore Sigma) lysis buffer supplemented with lysozyme (final concentration of 0.2 mg/ml), Benzonase (final amount of 50 units/ml), and protease inhibitor cocktails (1 tablet

per 10 ml lysate) as described in the manufacturer's instructions. Cleared lysates were loaded on chromatographic columns filed with His-select resin (Millipore Sigma) or glutathione resin (GE healthcare), respectively. Washes were performed for nickel resin purification using 20 column volumes of binding buffer (PBS containing 10 mM imidazole, pH 8.0) and 20 column volumes of wash buffer (PBS containing 30 mM imidazole, pH 8.0). For purification using glutathione resin, columns were washed using 20 column columns of PBS and eluted with PBS containing 10 mM reduced glutathione. Proteins were concentrated to 2 mL and further purified by size-exclusion chromatography using HiLoad 16/600 Superdex 200 (GE healthcare).

**Limited trypsinization**. TgIST-T2 that was co-purified with STAT1cc (10 μg) was digested at 25 °C for 5, 10, or 15 min by adding serial diluted trypsin (1 mg/mL stock) to 1:20, 1:40, 1:80, 1:160, 1:320, 1:640, 1:1280. Reactions were stopped by adding SDS sample buffer, followed by separation on SDS-PAGE gels of 12% or 15% acrylamide. Resistant bands treated with 1:160 diluted trypsin were cut from the gel, then reduced with 2 mM DTT, alkylated in 10 mM iodoacetamide for 20 min at 22 °C, and washed with 10 mM ammonium bicarbonate/acetonitrile. Trypsin was added and the digestion was carried out overnight at 37 °C. Peptides were extracted from the gel pieces, dried down, and re-dissolved in 2.5% acetonitrile, 0.1% formic acid. Finally, each digest was subjected to MS/MS analysis.

**Flow cytometry analysis**. RAW264.7 macrophages were either left uninfected or infected with parasites (MOI 3 to 1) for 6 h followed by activation with 100 U/mL IFN-γ for the final 18 h. Cells were collected by gentle scraping in cold PBS lacking cations and Fc receptors were blocked by TruStain FcX™ PLUS (anti-mouse CD16/32) antibody (BioLegend). Cells were then immunolabelled (1:100) with PE/Cyanine7 anti-mouse I-A/I-E antibody or a corresponding isotype control (BioLegend) in FACS buffer (1% BSA, 0.1% sodium azide in PBS). After staining, cells were fixed with ice-cold 4% formaldehyde and stored at 4 °C in PBS before analysis with SONY SH800 cell sorter. Data were analyzed and visualized using FlowJo.

**Expression and purification of STAT1 homodimer**. The strategy for purifying phosphorylated STAT1 was adopted from that previously described for STAT3[45] using the TKB1 strain that also expresses the tyrosine kinase Elk. Plasmid pET15b encoding the STAT1 core domain (residues S132-H713)[25] with a N-terminal Strep-tag was transformed into the E. coli TKB1 strain and expression was induced with 0.2 mM IPTG for 15 h at 16 °C. After induction, cells were harvested and resuspended in tyrosine kinase induction medium (M9 medium supplemented with 1 mM MgSO₄, 11 mM Glucose, 0.1% (w/v) casamino acids, 1.5 μM thiamine-HCl and 53 μM indole-acrylic acid) and grown for additional 2.5 h at 37 °C to facilitate phosphorylation of STAT1. Pelleted cells were lysed using Cellytic B lysis buffer with 20 mM DTT added, followed by purification using StreptactinXT resin (IBA Lifesciences). Eluted fractions were alkylated by adding 20 mM N-Ethylmaleimide (NEM, Sigma) to avoid protein aggregation due to cysteine crosslinking at high concentration[46]. The alkylation reaction was stopped by adding 50 mM β-mercaptoethanol. Protein samples were concentrated using Amicon Ultra-15 concentrators and further separated by HiTrap Heparin HP column and Hi-load 16/600 superdex 200 column (GE Healthcare). Fractions from the column were pooled and subjected for SEC-MALS analysis to determine their oligomeric status. STAT1 monomer was purified using the same construct similarly induced in the E. coli Rosetta strain, followed by the same purification strategy.

**Size exclusion chromatography and multi-angle light scattering (SEC-MALS)**. SEC-MALS experiments were carried out with a DAWN HELEOS II detector (Wyatt Technology) with an in-line size exclusion Superdex 200 (10/300 GL) column (GE Healthcare) balanced in 10 mM phosphate, pH 8.0, 137 mM NaCl, 2.7 mM KCl, 2 mM DTT at room temperature. Samples were injected into the column at 1.5 mg/mL in 100 μl total volume and molecular weights were determined using Astra 6 software (Wyatt Technology).

**Electrophoretic mobility shift (EMSA) assay**. IRDye labeled (5′ end) sense and anti-sense GAS (sense sequence: 5′-GATGTATTTCCCAGAAAAGG-3′ found upstream of the Fc-Gamma Receptor gene[47]), and ISRE (sense sequence: 5′-GGG AAAGGGAAACCGAAACTGAAGCC-3′, found upstream of the ISG15 gene[48]), DNA oligonucleotides were synthesized and annealed by Integrated DNA Technologies. DNA probes (0.05 pmole dsDNA per 20 μl reaction volume) were combined ±60 pmole STAT1 dimer in a total 20 μl reaction volume (EMSA buffer: 10 mM Tris pH 7.5, 50 mM KCl, 10 mM DTT, 0.125% Tween-20, 0.05 μg poly (dI·dC), 0.025 μg salmon sperm DNA) for 10 min at room temperature. Purified TgIST proteins were added to STAT1-DNA complexes and incubated for 10 min. Samples were mixed with LICOR 10X orange dye, resolved by 5% TBE gel (Bio-Rad), and visualized using an Odyssey infrared imager (LI-COR Biosciences).

**Bio-layer interferometry (BLI) assays**. The binding profile and apparent binding affinity of TgIST-T2 and STAT1 dimer were measured by an BLI using Octet-Red96 instrument (ForteBio). For some experiments, His-tagged STAT1 monomer or phosphorylated STAT1 dimer (pSTAT1d) were loaded onto Ni-NTA

biosensors. Alternatively, GST-tagged TgIST-R2 or various mutants were loaded onto anti-GST biosensors. Experiments were carried in kinetic buffer (ForteBio) at 25 °C. Data were analyzed and the binding curves were fit using the Data Analysis 9.0 software package (ForteBio). Association and dissociation curves were exported using Excel format and imported into Prism (GraphPad software) for visualization of the sensorgram.

**Circular dichroism**. All protein samples were prepared at a concentration of 100 μg/mL in 8 mM $Na_2HPO_4$, 1.5 mM $KH_2PO_4$, pH 7.8. CD spectra were acquired in duplicate using a Chirascan CD spectrometer (Applied Photophysics) and a 10 mm pathlength cuvette. After a one-minute incubation, wavelength scans from 200 to 280 nm were performed at 25 °C using a 20 nm/min scan rate. Signal from the buffer alone was subtracted.

**Protein crystallization**. Double-stranded GAS DNA oligonucleotides (sense sequence: 5'-ACAGTTTCCCGTAAATGC-3'[25]), were synthesized and annealed for crystallization (Integrated DNA Technologies). Phosphorylated STAT1 dimer or STAT1-linker-R2 was purified from TBK1 cells as described above and concentrated to 9.5 mg/mL. Protein-DNA complexes were obtained by mixing phosphorylated STAT1 dimer with GAS dsDNA at a molar ratio of 1:1.1 for 10 min at room temperature. Purified TgIST-R2 (G365-S420) protein containing a 6X His tag at the N-terminus was added into the STAT1-DNA mixture at a molar ration at 1:2.1 (STAT1 dimer: TgIST-R2) and incubated for 10 min. Crystals were obtained using the hanging drop vapor diffusion method by mixing 1 μl protein-DNA complex and 1 μl reservoir buffer (0.1 M Bis-Tris, pH 6.3, 3 M NaCl, 20 °C for STAT1 GAS-TgIST-R2 complex; 0.1 M sodium acetate pH 5.0, 0.1 M KCl, 0.02 M MgCl₂ and 20% PEG400, 20 °C for STAT1-GAS complex; 0.1 M Bis-Tris pH 6.3-6.0, 3 M sodium chloride and 50 mM sodium citrate tribasic, 20 °C for STAT1-linker-R2).

**Data collection and structure determination**. Data were collected at the Advanced Photon Source Beamline 19ID and Advanced Light Source (beamline 4.2.2). All three different crystallization conditions resulted in crystals that belong to same space groups C222₁. Diffraction data were indexed, integrated, scaled, and merged using iMOSFLM[49] or XDS[50]. The original STAT1 dimer complexed with DNA (PDB: 1BF5) was used as a molecular replacement search template by Phaser in CCP4. The refined STAT1 dimer structure was then used as the template for molecular replacement for the STAT1-TgIST-T2 complex. Structures were manually built in COOT[51] and then refined with Refmac5[52] or Phenix[53]. The combined structure was assessed by MolProbity[54] and visualized by PyMol[42]. Data collection and refinement statistics are listed in Supplementary Table 1.

**Reporting summary**. Further information on research design is available in the Nature Research Reporting Summary linked to this article.

## Data availability
The flow cytometry data have been deposited in FlowRepository under the accession code: FR-FCM-Z5FE [flowrepository.org/id/FR-FCM-Z5FE]. The crystal structure of phosphorylated STAT1 dimer complexed with repeat region from *Toxoplasma* protein TgIST have been deposited in Protein Data Bank with PDB ID: 8D3F. Other published structures, including STAT1 (PDB: 1BF5), monomeric STAT1 (PDB: 1YVL), STAT3 (PDB: 1BG1) and STAT6 (PDB: 5D39) can be found in Protein Data Bank. The proteomic data have been deposited in MassIVE under accession code MSV000089636. Sequences of human STAT1 were retrieved from the Uniprot database with accession number P42224; mouse, P42225; rat, Q9QXK0; pig, Q764M5; gorilla, G3SFV1; bovine, A0A3Q1ME65; cat, A0A337SVH3; chicken, Q5ZJK3; chimpanzee, A0A2I3TNY5; horse, A0A3Q2L3I5; and dog, A0A5F4C2J9. Source data are provided with this paper.

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

## Acknowledgements

We thank Daved Freemont for helpful advice, Gary Stormo for the locked C-C STAT1 dimer construct, and Jennifer Barks for technical support. This work was supported in part by a grant from the NIH to L.D.S. (AI118426), G.K.A. (P01AI120943), and the American Heart Association (AHA 831566/21) to Z.H.

## Author contributions

Conceptualization, Z.H., H.L., G.K.A., and L.D.S.; Data curation, Z.H. and J.N.; Formal analysis, Z.H., H.L., J.N., and C.K.; Funding acquisition, Z.H., G.K.A., and L.D.S.; Investigation, Z.H., J.N., C.K., and R.X.; Methodology, Z.H., J.N., and C.K.; Project administration, G.B., G.K.A., and L.D.S.; Visualization, Z.H. and H.L.; Writing Z.H., H.L., and L.D.S, Revisions, all authors.

## Competing interests

The authors declare no competing interests.
