## [Peer Review File · Nature Communications]

The intrinsically disordered protein TgIST from *Toxoplasma gondii* inhibits STAT1 signaling by blocking cofactor recruitmentREVIEWER COMMENTS

Reviewer #1 (Toxoplasma immunity, innate/adaptive immune responses) (Remarks to the Author):

NCOMMS-21-28570-T

In this study, Huang et al. used a variety of different techniques to examine the molecular basis for inhibition of transcription of genes downstream of type I/II IFN signaling by the *Toxoplasma gondii* secreted protein TgIST. First, they identified the region of the TgIST molecule that is required for this effect, followed by biophysical and structural analysis to demonstrate that the same region binds to STAT1 dimer. The final piece of data shows that in the presence of TgIST, STAT1 interacts less efficiently with the transcriptional CBP/p300 co-activator. The work is an extension of their published study that showed that TgIST inhibits STAT1 function by recruiting the Mi-2 Nucleosome Remodeling and Deacetylase (NuRD) complex, a chromatin-modifying repressor (Cell Host Microbe, 2016).

Although focused on one parasite (toxoplasma) protein, this work will be of interest to a wide readership of immunologist (and crystallographers) as it reveals new insights into STAT1 as well as the structure of the complex between TgIST fragment and STAT1 dimer + GAS dsDNA.

Major points:

1. The experiment shown in Figure 1 should include IP of U3A-STAT1 cells infected with the parasite but not treated with IFN-gamma to better support their finding, (made in the transfection assay using HEK293 cells) that TgIST binds only pSTAT1.

2. To assess the biological significance of TgIST inhibitory effect on STAT1 function, the authors should measure the level of MHC class I surface expression on U3A-STAT1 cells infected and uninfected +/- treated or not with IFN-gamma.

3. That TgIST associates with members of the Mi2/NuRD complex independently of STAT1 and IFN-gamma activation has been already shown and described in the Cell Host Microbe paper in 2016. The novelty of this work is that it demonstrates that internal repeats of TgIST interact with STAT1 while the entire molecule is required for interaction with Mi-2/NuRD complex. Based on the latter observation, the focus of the manuscript (after Fig. 2) shifts to the interaction between TgIST repeats and STAT1 dimer. First, the authors showed that in an in vitro transfection system this interaction is sufficient to prevent IRF-1 activation after 6 hr IFN-gamma stimulation. My concern regarding this experiment is the data shown in Supp. Fig. S2a (middle column). The expression of STAT1 in cells that are positive for GFP-TgIST appears to be lower than in cells that do not express TgIST. The intensity of the STAT1 signal needs to be quantified in TgIST positive and negative cells. The WB in F S2b is not sensitive enough to pick up differences.

4. In this work, the authors describe a new way that TgIST can inhibit STAT-1 function distinct from recruiting Mi-2/NuRD complex. While both pathways of inhibition are operative in HEK293 transfectants, it would be important to evaluate both in vivo. Can the TgIST-KO parasite be complemented with TgITS that either binds STAT1 or recruits Mi-2/NuRD complex?

5. Alternatively, is it possible/feasible to perform CHIP-Seq with STAT1 and or CBP on cells that were treated with the whole vs. the repeats-containing fragment of TgIST in order to better define the specific set of IFN-gamma stimulated genes (e.g. IRF-1/3 vs MHC class II Ag) in two conditions? The discussion about how different pathways of STAT1 suppression by TGIST may have different effects on primary vs. secondary response genes downstream of STAT-1 needs more elaboration in the discussion.

As a cellular immunologist, I don't feel qualified enough to comment on the second, structural part of the manuscript.

Minor comments:

No information about the parasite strain employed with Ty-tagged is provided in the corresponding Material and Methods section. What was the MOI of culture?

Line 148 ... has an extra word: allow

F2g panel needs an insert that will explain the abbreviations on the X-axis of the graph, as well as Fig 6F panel.

Reviewer #2 (Structural analyses, immunity against pathogens) (Remarks to the Author):

In this study, Huang et al. use a combination of immunoprecipitations (IPs), cell-based assays, immunofluorescence microscopy and biochemical assays to identify tandem repeat sequences within the effector protein TgIST from *Toxoplasma gondii* that are responsible for binding and transcriptional repression of STAT1. The authors use site-directed mutagenesis, X-ray crystallography, and biophysical assays to shed light on the molecular mechanism of transcriptional repression. IPs are first used to establish that TgIST binds STAT1 independently of the Mi-2/NuRD complex. The STAT1 binding region of TgIST was then mapped to within an 80-aa tandem repeat sequence located in the middle of the open reading frame. Deletion of one repeat sequence reduced STAT1 binding and IFN-gamma-dependent IRF1 expression, and deletion of both repeat sequences completely abolished STAT1 signaling. The authors then confirmed direct binding between the TgIST repeats and STAT1 using surface plasmon resonance, gel-shift assays and SEC-MALS using purified recombinant proteins. Residues required for binding were identified in a 7-aa core sequence within the repeat sequence. The K_d was found to be in the low nanomolar range. A crystal structure of TgIST Repeat 2 (R2) in complex with the STAT1 dimer showed the protein-protein interaction surface in detail, revealing an alpha-helical conformation for R2, which is mostly disordered in isolation, as shown by the authors' CD measurements. The authors ascribe changes in loop structures in STAT1 induced by TgIST binding as the reason for the failure to recruit CBP/p300, a claim supported by competition between STAT1 and CBP/p300 binding to TgIST R2.

Inhibition of STAT1 signaling is an important effector activity of the *T. gondii* parasite and the molecular mechanism of this activity has not been understood. The study builds on previous work by Sibley and colleagues showing that TgIST represses STAT1-dependent transcription and signaling. The present study provides important and novel insights on how the tandem

repeats from TgIST bind STAT1 and alter its structure. The authors propose a plausible model whereby the structural change induced by TgIST binding inhibit recruitment of CBP/p300. The study uses a complementary set of approaches and is well executed and logically presented, overall. A few concerns remain, as detailed below.

Concerns and comments (listed from most to least concerning)

1. Lines 258-275. Although the R2 alpha-helix has an amphipathic character, the claim that the amphipathic nature of the helix is important for STAT1 binding is potentially misleading or incorrect. From the structure it is apparent that the R2 peptide forms highly specific contacts with STAT1, including hydrophobic contacts and hydrogen bonds. Mutations at this interface such as V379R or L389E can be expected to disrupt binding because they introduce bulky and charged side chains at points where hydrophobic contacts are needed. Similarly, the R380V, E388L and D384A mutations would be expected to be disruptive because they remove hydrogen bonding potential. If, as the authors claim, it is the amphipathic nature of the helix that is important, mutations that preserve the amphipathic nature of the helix would not be expected to disrupt binding. The L383A mutation is disruptive, however, despite preserving the amphipathic nature of binding. Similarly, mutations such as R380E and E388R that preserve hydrophilicity but change polarity are likely to be disruptive (though these were not tested). This indicates that the amphipathic nature of the helix is incidental or a coincidence, and it is the specific details of the protein-protein interaction that are important. Hence, the authors should remove claims that the amphipathic nature of the R2 helix is important and focus on describing the interface as a specific protein-protein interaction, with its hydrophobic and hydrophilic components.
2. The authors zero in on the R2 sequence of TgIST as the key STAT1-binding region due to loss of repression activity in the Δ R2 TgIST mutant. It seems that a Δ R1 mutant was not tested for STAT1 binding and repression however, and it could be that such a mutant would have a similar phenotype to the Δ R2 mutant. It's unclear from the sequence of R1 why it wouldn't bind to STAT1. The authors should either provide support for a more important role for Δ R2 by testing the Δ R1 mutant or alter the text to leave more open the possibility that the two repeats are potentially equally important.
3. Lines 375-379. Based on the TgIST-STAT1 crystal structure and the TgIST protein sequence, could the R1 and R2 motifs from a single TgIST molecule in principle engage with both protomers of a STAT1 dimer? If both repeats could bind simultaneously this would be significant as it would increase the binding avidity.
4. Various mentions are made of predicted disorder and secondary structure for TgIST. The best predictions by far are now provided by AlphaFold 2. The authors should comment on the structure/disorder predictions for TgIST from AlphaFold 2. Ideally the authors should use the publicly available colab server to generate a prediction of the STAT1-TgIST-R1-R2 complex trying 2:2 and 2:1 stoichiometries.
5. Lines 146-147. It should be clarified that the T4 construct contains only the repeats without any NLS or signal peptide (according to Fig. 1d). Or perhaps the authors meant T2 instead of T4? Indeed, T2 seems like a fairer comparison to T2 Δ R2 than T4.
6. Lines 234-236. The wording of this sentence is unclear and potentially confusing- please clarify.

7. Fig. 4d. The authors should consider adding a cartoon outline of the bound TgIST peptide (in a light or transparent color) to help orient the reader.

8. Line 313, and throughout. When citing a PDB entry the authors should also include the primary paper citation.

9. There are many typos and grammatical errors throughout the text, which needs additional proofreading by a native English speaker. For example:

(i) Line 31. Replace “unexpectedly” with “unexpectedly”.

(ii) Line 42. Change “transportation” to “transport”.

(iii) Line 66. Change “disorder” to “disordered”.

(iv) Line 148. Change “allow” to “although”.

(v) Line 191. Change “multiple angle” to “multiangle”

(vi) Line 310. Change to “...competes with the interaction...:

(vii) Line 339. Change “bind” to “binding”.

(viii) Line 365. Change “exportation” to “export”.

(ix) Line 528. Change “TgIST-R1-bound” to TgIST-R2-bound”.

REVIEWER COMMENTS

Reviewer #1 (Toxoplasma immunity, innate/adaptive immune responses) (Remarks to the Author):

NCOMMS-21-28570-T

In this study, Huang et al. used a variety of different techniques to examine the molecular basis for inhibition of transcription of genes downstream of type I/II IFN signaling by the *Toxoplasma gondii* secreted protein TgIST. First, they identified the region of the TgIST molecule that is required for this effect, followed by biophysical and structural analysis to demonstrate that the same region binds to STAT1 dimer. The final piece of data shows that in the presence of TgIST, STAT1 interacts less efficiently with the transcriptional CBP/p300 co-activator. The work is an extension of their published study that showed that TgIST inhibits STAT1 function by recruiting the Mi-2 Nucleosome Remodeling and Deacetylase (NuRD) complex, a chromatin-modifying repressor (Cell Host Microbe, 2016).

Although focused on one parasite (*Toxoplasma*) protein, this work will be of interest to a wide readership of immunologist (and crystallographers) as it reveals new insights into STAT1 as well as the structure of the complex between TgIST fragment and STAT1 dimer + GAS dsDNA.

Major points:

1. The experiment shown in Figure 1 should include IP of U3A-STAT1 cells infected with the parasite but not treated with IFN-gamma to better support their finding, (made in the transfection assay using HEK293 cells) that TgIST binds only pSTAT1.

>We appreciate the suggestion and have added a blot that shows these results to the supplementary materials (Fig. S1). The secreted TgIST was able to pull down STAT1 in IFN- γ treated U3A-STAT1 cells (Fig 1a), but not in non-IFN- γ treated cells (new Fig. S1), further supporting the conclusion that TgIST interacts only with the phosphorylated STAT1 dimer. Please refer to lines 97-101 and Fig. S1.

2. To assess the biological significance of TgIST inhibitory effect on STAT1 function, the authors should measure the level of MHC class I surface expression on U3A-STAT1 cells infected and uninfected +/- treated or not with IFN-gamma.

>We appreciate the suggestion on studying the biological role of TgIST in STAT1 signaling, especially on genes that are induced downstream of IRF-1 that involve immune responses. We generated a new Fig. 3 to address these questions, both *in vitro* and *in vivo* (also proposed by this reviewer in Major point 4). Please refer to lines 211-236.

>In terms of MHCI expression, we felt that testing this on U3A cells might not be the best system since they overexpress STAT1. Therefore we measured the surface expression of MHCI on immune cells using RAW246.7 cells by flow cytometry. Briefly, MHCI expression was indeed down-regulated by *T. gondii* infection, and we also found this regulation was dependent on the virulence effector TgIST. Moreover, the N-terminal region of TgIST bearing the repeat regions that binding STAT1 was sufficient to down

regulate MHCI, independent of the C-terminal Mi-2/NuRD binding region. These new data are found in Fig. 3. Please refer to lines 211-236 and Figure 3.

3. That TgIST associates with members of the Mi2/NuRD complex independently of STAT1 and IFN-gamma activation has been already shown and described in the Cell Host Microbe paper in 2016. The novelty of this work is that it demonstrates that internal repeats of TgIST interact with STAT1 while the entire molecule is required for interaction with Mi-2/NuRD complex. Based on the latter observation, the focus of the manuscript (after Fig. 2) shifts to the interaction between TgIST repeats and STAT1 dimer. First, the authors showed that in an in vitro transfection system this interaction is sufficient to prevent IRF-1 activation after 6 hr IFN-gamma stimulation. My concern regarding this experiment is the data shown in Supp. Fig. S2a (middle column). The expression of STAT1 in cells that are positive for GFP-TgIST appears to be lower than in cells that do not express TgIST. The intensity of the STAT1 signal needs to be quantified in TgIST positive and negative cells. The WB in F S2b is not sensitive enough to pick up differences.

>We agree there is some diminished signal for STAT1 in cells over-expressing TgIST. However, this lower level is seen in all samples expressing different constructs and so it is not the basis for the differences in ability to block IRF1 induction. We have added a quantitative comparison to Fig. S3b (previously Fig. S2) to show that the level of STAT1 is similar among the different TgIST constructs being compared. Please refer to lines 186 – 197 and supplemental Fig S3b.

4. In this work, the authors describe a new way that TgIST can inhibit STAT-1 function distinct from recruiting Mi-2/NuRD complex. While both pathways of inhibition are operative in HEK293 transfectants, it would be important to evaluate both in vivo. Can the TgIST-KO parasite be complemented with TgITS that either binds STAT1 or recruits Mi-2/NuRD complex?

>We appreciate this suggestion and agree that this is an interesting extension of the work. In new studies, we monitored the propagation of transgenic parasites in vivo using bioluminescent imaging. We have generated several transgenic parasite lines bearing either wildtype TgIST or TgIST-T1 version that only binds to STAT1 but not to Mi-2/NuRD complex. We have tested the dissemination of these parasites in vivo and provided new data in Fig. 3. In brief, we found that parasites expressing TgIST-T1 that lacks Mi-2/NuRD binding can still inhibit IRF1 and MHCI expression in vitro (as we mentioned in comments reply question #2). More importantly, TgIST-KO parasite complemented with TgIST-T1 can also restore parasite dissemination to the wildtype level. These findings demonstrate that new mechanism we have defined for how inhibits STAT1 signaling in vitro is also functional in the mouse infection model. These in vitro and in vivo data provide further support for the importance of structural studies in the manuscript as these studies provide a molecular mechanism for the observed block in STAT1 transcription. Please refer to lines 237-251 and Figure 3.

5. Alternatively, is it possible/feasible to perform ChIP-Seq with STAT1 and or CBP on cells that were treated with the whole vs. the repeats-containing fragment of TgIST in order to better define the specific set of IFN-gamma stimulated genes (e.g. IRF-1/3 vs MHC class II Ag) in two conditions? The discussion about how different pathways of STAT1 suppression by TgIST may have different effects on primary vs. secondary response genes downstream of STAT-1 needs more elaboration in the discussion.

>We agree that this would be an interesting extension of the study, but feel it is outside the scope of the present work. As it currently stands, the manuscript is oversized relative to the journal specifications. If we were to add a ChIP-Seq data, it would either have to replace some existing material or it would have to be relegated to the supplement, neither of which would be optimal. Our preference would be to delay these studies until we can further define the NuRD binding region and then explore the set of genes that are differentially regulated by the entire TgIST protein, the N terminal domain that binds STAT1 through the repeat region, and the C terminal NuRD binding region. Potential differences in the roles of these domains are summarized in paragraph 2 of the Discussion. Please refer to lines 431-443.

As a cellular immunologist, I don't feel qualified enough to comment on the second, structural part of the manuscript.

Minor comments:

No information about the parasite strain employed with Ty-tagged is provided in the corresponding Material and Methods section. What was the MOI of culture?

> We have updated these details in the manuscript. Please refer to lines 694-701.

Line 148 ... has an extra word: allow

> Corrected.

F2g panel needs an insert that will explain the abbreviations on the X-axis of the graph, as well as Fig 6F panel.

> We have updated the legend to include these details. Please refer to lines 558-559, 675.

Reviewer #2 (Structural analyses, immunity against pathogens) (Remarks to the Author):

In this study, Huang et al. use a combination of immunoprecipitations (IPs), cell-based assays, immunofluorescence microscopy and biochemical assays to identify tandem repeat sequences within the effector protein TgIST from *Toxoplasma gondii* that are responsible for binding and transcriptional repression of STAT1. The authors use site-directed mutagenesis, X-ray crystallography, and biophysical assays to shed light on the molecular mechanism of transcriptional repression. IPs are first used to establish that TgIST binds STAT1 independently of the Mi-2/NuRD complex. The STAT1 binding region of TgIST was then mapped to within an 80-aa tandem repeat sequence located in the middle of the open reading frame. Deletion of one repeat sequence reduced STAT1 binding and IFN-gamma-dependent IRF1 expression, and deletion of both repeat sequences completely abolished STAT1 signaling. The authors then confirmed direct binding between the TgIST repeats and STAT1 using surface plasmon resonance, gel-shift assays and SEC-MALS using purified recombinant proteins. Residues required for binding were identified in a 7-aa core sequence within the repeat sequence. The K_d was found to be in the low nanomolar range. A crystal structure of TgIST Repeat 2 (R2) in complex with the STAT1 dimer showed the protein-protein interaction surface in detail, revealing an alpha-helical conformation for R2, which is mostly disordered in isolation, as shown by the authors' CD measurements. The authors ascribe changes in loop structures in STAT1 induced by TgIST binding as the reason for the failure to recruit CBP/p300, a claim supported by competition between STAT1 and CBP/p300 binding to TgIST R2.

Inhibition of STAT1 signaling is an important effector activity of the *T. gondii* parasite and the molecular mechanism of this activity has not been understood. The study builds on previous work by Sibley and colleagues showing that TgIST represses STAT1-dependent transcription and signaling. The present study provides important and novel insights on how the tandem repeats from TgIST bind STAT1 and alter its structure. The authors propose a plausible model whereby the structural change induced by TgIST binding inhibit recruitment of CBP/p300. The study uses a complementary set of approaches and is well executed and logically presented, overall. A few concerns remain, as detailed below.

Concerns and comments (listed from most to least concerning)

1. Lines 258-275. Although the R2 alpha-helix has an amphipathic character, the claim that the amphipathic nature of the helix is important for STAT1 binding is potentially misleading or incorrect. From the structure it is apparent that the R2 peptide forms highly specific contacts with STAT1, including hydrophobic contacts and hydrogen bonds. Mutations at this interface such as V379R or L389E can be expected to disrupt binding because they introduce bulky and charged side chains at points where hydrophobic contacts are needed. Similarly, the R380V, E388L and D384A mutations would be expected to be disruptive because they remove hydrogen bonding potential. If, as the authors claim, it is the amphipathic nature of the helix that is important, mutations that preserve the amphipathic nature of the helix would not be expected to disrupt binding. The L383A mutation is disruptive, however, despite preserving the amphipathic nature of binding. Similarly, mutations such as R380E and E388R that preserve hydrophilicity but change polarity are likely to be disruptive (though these were not tested). This indicates that the amphipathic nature of the helix is incidental or a coincidence, and it is the specific details of the protein-protein interaction that are important. Hence, the authors should remove claims that the amphipathic nature of the R2 helix is important and focus on describing the interface as a specific protein-protein interaction, with its hydrophobic and hydrophilic components.

>We agree with the reviewer that the interaction between TgIST and the binding site on STAT1 involves multiple interactions that include electrostatic and hydrophobic contacts. We also agree that the

mutations we have made might disrupt binding due to interruption of some of these features individually, or due to larger structural changes. The finding that TgIST adopts a helical conformation when bound to STAT1 is notable given its disordered state when unbound. For this reason, we considered that one of the properties that is important for binding is the amphipathic helix as it presents appropriate charged and hydrophobic interaction at key points in the interface. Altering those contact sites clearly can have an effect independent of the helical nature, but without the secondary structure, they would also be misplaced. We have modified the text to provide a more balanced summary of the importance of these different features of the binding interface between TgIST and STAT1. Please refer to lines 338-342, 352-356.

2. The authors zero in on the R2 sequence of TgIST as the key STAT1-binding region due to loss of repression activity in the Δ R2 TgIST mutant. It seems that a Δ R1 mutant was not tested for STAT1 binding and repression however, and it could be that such a mutant would have a similar phenotype to the Δ R2 mutant. It's unclear from the sequence of R1 why it wouldn't bind to STAT1. The authors should either provide support for a more important role for Δ R2 by testing the Δ R1 mutant or alter the text to leave more open the possibility that the two repeats are potentially equally important.

>We agree that R1 is highly similar and it is also capable of binding to STAT1. Our data from biochemical and cell-based assays both demonstrate that R1 binds to STAT1 (see Fig. 2b and 2c, construct TgIST-T2 Δ R2, which only bears R1 but can still bind the STAT1 dimer and block IRF1 expression). We have included both sequenced that are aligned in Fig. S4b to highlight that it shares the core motif that is critical for STAT1 binding. We have also modified the text to suggest both may play a role in STAT1 binding. Please refer to lines 166-168, 194-195, and Fig. S4b)

3. Lines 375-379. Based on the TgIST-STAT1 crystal structure and the TgIST protein sequence, could the R1 and R2 motifs from a single TgIST molecule in principle engage with both protomers of a STAT1 dimer? If both repeats could bind simultaneously this would be significant as it would increase the binding avidity.

>This is an excellent suggestion, and it fits the BLI data presented in Fig. 4. Cell based assay in Fig. 2g also suggested two repeats containing TgIST bound STAT1 more efficiently. We have modified the text to include this possibility in the Discussion. Please refer to lines 462-465.

4. Various mentions are made of predicted disorder and secondary structure for TgIST. The best predictions by far are now provided by AlphaFold 2. The authors should comment on the structure/disorder predictions for TgIST from AlphaFold 2. Ideally the authors should use the publicly available colab server to generate a prediction of the STAT1-TgIST-R1-R2 complex trying 2:2 and 2:1 stoichiometries.

>We added the secondary structure prediction result using AlphaFold 2 in Fig. S8e. All predictions across different tools gave a helical prediction when using TgIST-R2 as the input. However, these results are 5-6 amino acid longer than the structure we saw from the crystal structure. We have also generated a model of the TgIST-STAT1 interaction using AlphaFold 2 and unfortunately it does not do a good job of predicting the structure of this complex. The colab server for AlphaFold 2 scripts was used to predict the complex structure using protein sequences with different stoichiometries: STAT1-TgIST-R2

(crystallographic structure solved in this paper in Fig. a, prediction in Fig. b), STAT1:R2=2:1 (Fig. c) and STAT1:R2=2:2 (Fig. d). As mentioned above, we see the helical structure formed by the core binding domain of TgIST-R2, but the location of the peptide is different from the crystal structure. Instead of sitting on top of the STAT1 dimer interface, as we observed from our crystal structure, R2 relocates to the center of this dimer interface. This placement is not consistent with the actual structure and it also creates a steric hinderance with STAT1 as seen in panel b below. The incorrect placement of the TgIST peptide was also not corrected by adopting different stoichiometries as shown in c, d. Due to these inconsistencies, we decided not to include them in the paper but instead provide them here for the benefit of the review.

a. Crystal structure

b. AlphaFold prediction using STAT1-linker-R2

c. AlphaFold prediction using STAT1:RP2=2:1

d. AlphaFold prediction using STAT1:R2=2:2

Yellow: TgIST-R2

Green and cyan: STAT1 protomer in dimer

5. Lines 146-147. It should be clarified that the T4 construct contains only the repeats without any NLS

or signal peptide (according to Fig. 1d). Or perhaps the authors meant T2 instead of T4? Indeed, T2 seems like a fairer comparison to T2ΔR2 than T4.

>We clarified T4 construct in the parentheses. We also added T2 in the text to better address the fact that two internal repeats of TgIST are more efficient for binding to STAT1 dimer. Please refer to lines 558-559.

6. Lines 234-236. The wording of this sentence is unclear and potentially confusing- please clarify.

>We have revised the text to state: One STAT1 protomer shares a 551 Å² buried surface area with TgIST-R2, buried the helix deep into the groove. Please refer to lines 306-108.

7. Fig. 4d. The authors should consider adding a cartoon outline of the bound TgIST peptide (in a light or transparent color) to help orient the reader.

>We added the peptide in a transparent view in Fig 5d (previous Fig 4d).

8. Line 313, and throughout. When citing a PDB entry the authors should also include the primary paper citation.

>We have updated these details in the manuscript. Please refer to lines 371, 400.

9. There are many typos and grammatical errors throughout the text, which needs additional proofreading by a native English speaker.

>The follow typos and errors have been corrected and the manuscript has been carefully checked.

(i) Line 31. Replace “unexpectedly” with “unexpectedly”.

>Corrected

(ii) Line 42. Change “transportation” to “transport”.

>Corrected

(iii) Line 66. Change “disorder” to “disordered”.

>Corrected

(iv) Line 148. Change “allow” to “although”.

>Corrected

(v) Line 191. Change “multiple angle” to “multiangle”

>Corrected

(vi) Line 310. Change to “...competes with the interaction...:

>Corrected

(vii) Line 339. Change “bind” to “binding”.

>Corrected

(viii) Line 365. Change “exportation” to “export”.

>Corrected

(ix) Line 528. Change “TgIST-R1-bound” to TgIST-R2-bound”.

>Corrected

REVIEWERS' COMMENTS

Reviewer #1 (Remarks to the Author):

None

Reviewer #2 (Remarks to the Author):

All concerns from the first round of review have been adequately addressed.